# Precipitation disaster hotspots depend on historical climate variability

Iris de Vries [1,2] ✉, Maybritt Schillinger [3], Erich Fischer [1],
Sebastian Sippel [4] & Reto Knutti [1]

Record-high precipitation events are relevant for impacts since they are more severe than any observed event and can lead to unforeseeable consequences. Climate change increases average record-breaking probability, but the current, local record-breaking probability and local disaster preparedness are dependent on observed precipitation history as well. Here, we show that historical variability shapes current and future record-breaking probabilities: regions with low current records are more at risk. Climate change modifies this pattern non-linearly: moderate climate change (SSP2-4.5) increases average record-breaking probability by 2050 by 40%, but high current records are most sensitive to climate change with a record-breaking probability increase of up to 75%. Thus, regions with low current records are most at risk, but regions with high current records see the steepest risk increase with climate change. Disaster risk is further increased by low preparedness. If the last record-breaking event is long ago, local society is more likely to be unprepared for the next one. Vulnerability and exposure in many regions with high record-breaking probability is high due to poverty and rapid urbanisation, resulting in a major imminent threat.

Extreme rainfall contributes to natural disasters that incur enormous human and economic damage each year[1]. Many studies on the effect of climate change on extreme precipitation frequencies and intensities in both models and observations exist, and report (projected) increases over almost all the global land, in agreement with theoretical understanding of the physical processes involved[2–21]. However, the local weather we experience does not behave like a smooth climate change trend, but is predominantly determined by unforced natural variability, particularly for a highly variable phenomenon such as extreme precipitation. In this study, we integrate information on climate change and natural variability to estimate where near-term precipitation-caused disasters are more likely to occur.

The high variability of precipitation implies that events that are much more extreme than any previously observed event can occur, catching society by surprise. Record-breaking and so-called record-shattering events[22–24] – large jumps in record values–become less likely

as observational timeseries length increases, but remain possible due to natural variability even in a stationary climate. Climate change increases record-breaking and shattering weather event probabilities relative to a stationary climate[22–29], and the probability increase is particularly strong for record-breaking precipitation events due to the non-linear response of extreme precipitation to warming[23,30]. Theory, models and observations agree that, on average, extreme precipitation increases at a rate of ~7% per degree Celsius of global warming[3,13,31]. However, the rate of increase varies regionally[2,14,16], and rare extreme precipitation events see relatively stronger increases than more moderate ones for reasons that are yet to be fully understood[12,14,32,33]. This implies that extreme precipitation variability increases, and hence also the probability of large record jumps. The probability of experiencing local future record jumps in reality depends on the combination of the to-be-broken standing record and the changing probability distribution of extreme precipitation. Combining future projections

[1]Institute for Atmospheric and Climate Science, ETH Zürich, Zürich, Switzerland. [2]Department of Earth, Atmospheric and Planetary Sciences, MIT, Cambridge, MA, USA. [3]Seminar for Statistics, ETH Zürich, Zürich, Switzerland. [4]Leipzig Institute for Meteorology, Leipzig University, Leipzig, Germany. ✉e-mail: irisedevries@gmail.com

and historical observations of precipitation extremes can thus provide locally specific and useful estimates of future high-impact event probabilities.

The focus of this study is record-breaking annual daily maximum precipitation (Rx1d) events. Record breaking is our event type of choice, since record events lie, by definition, outside the historically observed range of intensities, and are therefore likely to cause damage and perhaps exceed the levels infrastructure and emergency plans are designed for. Of course, record-breaking events do not always have high impacts, and high impacts can also result from non-record-breaking events. While other definitions of extremes and high-impact events deserve equal attention, we focus here on record-breaking events, given their clear statistical and conceptual definition and their intuitive connection to unpredictable impacts. Record-breaking probabilities are commonly assessed in a marginal, history-unaware manner by averaging many climate realisations, aggregating in space, or determining marginal probability distributions[22–29]. We use a more experience-centric approach to record-breaking events, focused on the path-dependent probability of breaking the local observed standing record by conditioning probabilities on historical observations. A number of recent studies has explored similar questions using different approaches that acknowledge and make use of historical event information[34–36].

Precipitation history affects record-breaking and disaster probability in different ways. Natural variability can lead to extended periods of multiple decades with only relatively low (or high) levels of extreme precipitation[37–39]. It is thus likely that regions exist where the historical record event within the observational timeseries has a low intensity relative to the underlying Rx1d distribution, and/or where the last record-breaking event occurred a long time ago. Regions with low records are exposed to higher record-breaking probability, which can be further heightened over time by climate change. In addition, in regions where no salient, high-impact precipitation events occurred in recent history, extreme event probabilities might be underestimated, which may propagate into decision making processes at the local level, leading to lower disaster preparedness[40,41]. Cognitive biases such as availability bias distort our judgement of event probabilities based on the ease with which the events come to mind; salient and recent events come to mind easily, whereas mild and long-ago events do not[42]. Indeed, there are numerous anecdotal examples where long periods without record-breaking events or disasters (record gaps) plausibly led to reduced risk awareness and unpreparedness when disaster struck[43–48], and previous experience is highly correlated with preparedness[49]. In summary, a low standing record implies the probability of exceeding it is high, and absence of recent high-impact extremes can lead to low local risk awareness and preparedness despite quantitatively high extreme event probability.

Given the above considerations, the three main factors that we consider to be of relevance for local precipitation record-breaking with high disaster potential are 1) low historical record values, 2) strong climate change relative to natural variability, and 3) long historical record gaps. Ultimately, the goal of this paper is to make society and local governments aware that disaster potential is high in many world regions due to a combination of natural variability and climate change, exacerbated by inadequate preparedness due to societal and political judgement biases. Moving away from reactive, transient responses and towards sustainable, long-term disaster resilience requires us to accurately determine local natural hazard probabilities and risks, while identifying and correcting biases that impede the development and implementation of robust disaster policies.[47,48,50,51]

In this work, we assess local future Rx1d record-breaking probabilities conditional on historical observations, while accounting for non-stationarity in the distribution of extreme precipitation. We use two gridded station observation datasets – HadEX3[52] and REGEN[53] – and the ERA5 reanalysis[54] to determine current record levels based on the period 1950–2015. Climate change induced shifts in the record-

breaking probability are based on simulated Rx1d data from a multi-model ensemble of the Coupled Model Intercomparison Project Phase 6 (CMIP6) forced with moderate warming scenario SSP2-4.5[55,56]. In the following we show that future record-breaking probability estimates are strongly shaped by historical record levels, and modulated by climate change in a non-linear fashion. Combining historical record levels and record gaps allows us to determine which regions are at high risk for precipitation caused disaster, as well as the global number of people exposed to such high disaster potential now and in the future.

## Results
### Framework

As outlined in the introduction, future local record-breaking probabilities depend on the historical record and climate change. For four single gridcells with all possible combinations of low/high historical records and weak/strong climate change, Fig. 1 illustrates the analysis metrics we use throughout this paper. The analysis metrics refer to either marginal or conditional probabilities, in either stationary or non-stationary conditions. Marginal probabilities are history-unaware probabilities based on the mean climate, conditional probabilities are history-aware probabilities that depend on historical observations. Stationary metrics assume a counterfactual world without climate change, and non-stationary metrics include effects of anthropogenic climate changes. The four gridcells were selected to illustrate the effects of observed record extremeness and climate change. We selected gridcells with record quantile levels corresponding to the ~10th percentile and 90th percentile of all observed gridcells–0.97 and 0.999, respectively. Similarly, they feature either weak or strong climate change, defined as a relative change in Rx1d intensity from 2016 to 2100 approximately equal to the 10th percentile or the 90th percentile of all simulated Rx1d trends. Multiple gridcells meet these criteria; the final four were selected to represent a range of climatic and geographical regions. The geographical locations we refer to in Fig. 1 are located in the selected gridcells, the data shown are the gridcell values from HadEX3.

In the local Rx1d anomaly timeseries in Fig. 1a, the current record is marked and its quantile level relative to the gridcell-specific observational Rx1d probability density function is printed and indicated to the right of the timeseries. The quantile level reflects how extreme the record is. The quantile level of an Rx1d record of intensity $L$, occurring in year $T$ is $F_{T,\text{obs}}(L)$. Here, $F_{t,\text{obs}}(\text{Rx1d})$ refers to the cumulative distribution function (CDF) of the local non-stationary generalised extreme value distribution (GEV) fitted to the observations, dependent on time $t$. The non-stationarity of the GEV is needed to evaluate the extremeness of the record in its year of occurrence.

It is important to note that observational GEV distributions are subject to high uncertainty, especially in the tails. Therefore, also observed record quantile levels are uncertain, and records might erroneously be classified as particularly low or high. We quantify and show uncertainty ranges for all results, and emphasise that confidence in specific quantitative values is limited.

To calculate future record breaking probabilities, we first determine the temporal evolution of the observed record quantile level with respect to simulated climate change, $F_t(L_m)$. Here, $F_t(\text{Rx1d})$ is the CDF of Rx1d according to non-stationary GEV distributions of Rx1d simulated by CMIP6 models, and $L_m$ is the quantile-mapped climate model specific Rx1d level corresponding to the observed record quantile level. The bottom halves of the subplots in Fig. 1a show $F_t(L_m)$ from the record-setting year $T$ to 2100, where shading represents uncertainty ranges in observed quantile level (95% confidence interval, based on bootstrapped GEV coefficients) and model spread in projected Rx1d distributional changes. In the absence of climate change, $F_t(L_m)$ would be constant.

The local Rx1d record-breaking probability at time $t$ is simply $1 - F_t(L_m)$, and we denote it by $\Pr_t(R|L)$, see Equation (1). Here, $R$ is a

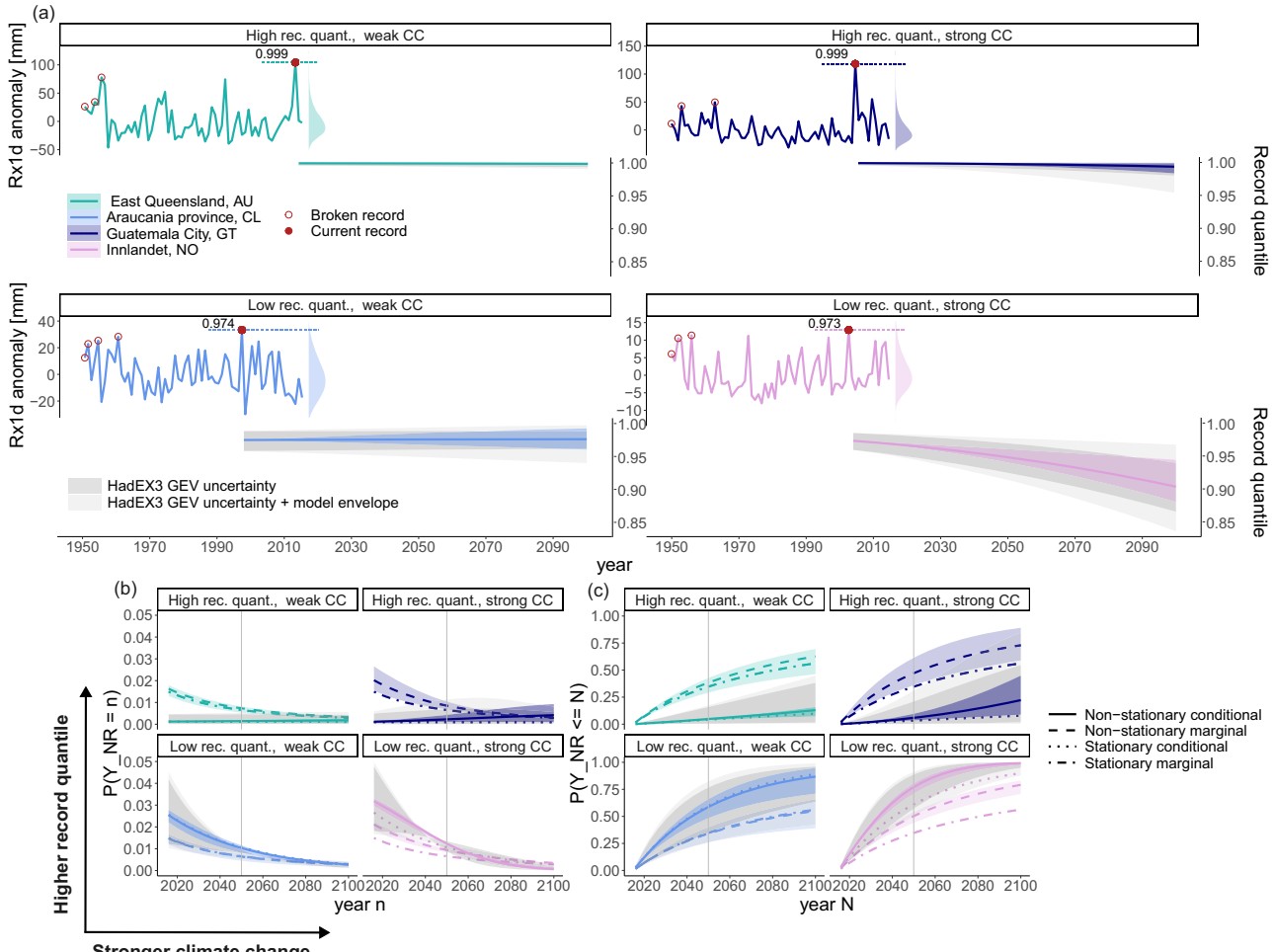

**Fig. 1 | Illustration of the record-breaking probability metrics used in this study, based on observed records in annual maximum daily precipitation (Rx1d) in HadEX3 at four locations which differ in historical record quantile level and climate change trend in Rx1d. a** Upper half: local annual Rx1d anomalies relative to the full-period 1950–2015 average, records (red markers), and observational distributions of Rx1d. Lower half: temporal evolution of the standing record's quantile level (right y axis) as projected by eight CMIP6 models forced with SSP2-4.5. **b** Pr($Y_{NR} = n$): the probability of setting the next Rx1d record in year $n$, starting from the end of observations in 2015. **c** Pr($Y_{NR} \leq N$): cumulative probability of breaking the Rx1d record in any year between 2016 (the first unobserved year) and $N$. In all plots, coloured shading shows the model mean envelope resulting from differences in Rx1d trends across different climate models, dark grey shading shows the multi-model mean uncertainty range resulting from observational generalised extreme value distribution (GEV) fit uncertainties (see Sect. 4.2), light grey shading encompasses the range of observational GEV uncertainties combined with across-model differences. Vertical grey lines indicate the year 2050.

binary variable indicating whether a record is broken (1) or not (0) and probability of record breaking is expressed conditional on the observed record level $L$.

$$\mathrm{Pr}_t(R|L) = \mathrm{Pr}(\mathrm{Rx1d}_t > L) = 1 - F_t(L_m) \qquad (1)$$

Thus, a decrease in $F_t(L_m)$ with climate change implies that conditional record-breaking probability increases. The shading shows that uncertainties in quantile levels and their evolution are largest for lower quantile levels, corresponding to larger uncertainties in the highest record-breaking probabilities. The implications of these uncertainties for all results are discussed in the sections below. Sect. 4 contains detailed descriptions of the GEV fitting, quantile mapping, and uncertainty quantification procedures.

Next, we compute the probability of the next record-breaking event to occur in year $n$. We denote it by Pr($Y_{NR} = n$): this refers to the probability of not breaking the record in any year between 2016 and year $n - 1$, and breaking it in year $n$.

Pr($Y_{NR} = n$) is different from the annual record-breaking probability in a single year $\mathrm{Pr}_t(R)$, as it represents the probability of a specific record-breaking evolution over multiple years. Pr($Y_{NR} = n|L$) is

given in equation (2) and shown in the solid lines in Fig. 1b for non-stationary Rx1d distributions. The dotted lines show the stationary reference case without climate change, where $\mathrm{Pr}_t(R|L)$ is constant with time. In case of a low historical record quantile level, Pr($Y_{NR} = n|L$) is high for early years and decreases over time, indicating that it is likely that the record is broken rather quickly (compare solid lines in bottom quadrants to top quadrants in Fig. 1b). Climate change increases Pr($Y_{NR} = n|L$), but its effect is small due to the generally high record-breaking probability. In case of a high historical record quantile level, Pr($Y_{NR} = n|L$) is generally small for all $n$, but more sensitive to climate change trends towards the end of the century (solid lines in top quadrants Fig. 1b). For low record quantile levels, uncertainties decrease with time as Pr($Y_{NR} = n|L$) approaches 0 (100% chance that the record has been broken already), whereas uncertainties increase for higher quantile levels due to model differences in climate change effects on record quantile level (see also Sect. 2.3).

$$\mathrm{Pr}(Y_{NR} = n|L, T = 2015) = \prod_{t=T+1}^{n-1} [1 - \mathrm{Pr}_t(R|L)] \cdot \mathrm{Pr}_n(R|L) \qquad (2)$$

Our final metric of interest, the conditional cumulative record-breaking probability (CCP), represents the probability of breaking the next record by some year $N$, given by equation (3). The cumulative summation of yearly values of $\Pr(Y_{NR} = n|L)$ is valid since Rx1d values and record-breaking probabilities in different years are independent (Supplementary Fig. S1). By substituting values of $\Pr(Y_{NR} = n|L)$ for a non-stationary and stationary climate, CCP metrics for a non-stationary and stationary climate are obtained, which we refer to as non-stationary and stationary CCP.

$$\Pr(Y_{NR} \leq N|L, T = 2015) = \sum_{n=T+1}^{N} \Pr(Y_{NR} = n|L) \qquad (3)$$

The effects of the historical record quantile level and climate change are evident in CCP, shown in Fig. 1c: low historical record quantile levels (bottom row) imply high $\Pr(Y_{NR} = n|L)$ values in the first few years after 2015 and thus an initially high and steeply increasing CCP. CCP converges to 1 for higher $N$, indicating a high chance that the current record will be broken in the near term. Stronger climate change amplifies the increase in CCP with $N$. For high historical record quantile levels, CCP is very low initially due to low $\Pr(Y_{NR} = n|L)$ values, but the effect of climate change becomes apparent over time. In CCP, the propagating effects of uncertainties in observational GEV fits (grey) and climate change across models (colour) are clearly visible. Even at locations with strong climate change, the uncertainties in record quantile levels brought about by observational GEV uncertainties dominate. This is in line with well known sensitivities of GEV fits to limited sample sizes[57-59].

In order to quantify the effect of conditioning on the observed record, we also determine marginal (unconditional) cumulative record-breaking probabilities (MCP). MCP reflects the probability aggregated over all quantile levels and is thus independent of specific local records; it is the result one would obtain if one could average over many independent Rx1d timeseries for the same location and period. In a non-stationary climate, unconditional $\Pr_t(R)$ is given by equation (4)[60], in which $f_t(Rx1d)$ and $F_t(Rx1d)$ are the probability density function (PDF) and CDF of non-stationary GEV distributions fitted to the simulated Rx1d data. The counterfactual stationary $\Pr_t(R)$ is independent of the underlying distribution: the record-breaking probability in the $j$th observation is simply $\frac{1}{j}$[61]. Here, $j$ is the timestep corresponding to year $t$; in our case $j = 1$ in 1950.

$$\Pr_t(R) = \Pr(Rx1d_t > \max(Rx1d_1, \dots Rx1d_{t-1}))$$
$$= \int_{-\infty}^{\infty} f_t(Rx1d) \prod_{i=1}^{t-1} F_i(Rx1d) \, dRx1d \qquad (4)$$

The result of equation (4) and $\frac{1}{j}$ are substituted in equation (2) to obtain non-stationary and stationary marginal $\Pr(Y_{NR} = n)$, respectively. These are then substituted into (3) without conditioning on $L$ to obtain non-stationary and stationary MCP. Marginal $\Pr(Y_{NR} = n)$ is shown in Fig. 1b, as dashed and dash-dot lines. As the marginal probability of not breaking the historical record between 2016 and $n$ decreases strongly as time passes (higher $n$), $\Pr(Y_{NR} = n)$ decreases with $n$ as well. Climate change modifies the decrease: stronger climate change implies higher chances the record is broken earlier, which can be seen by comparing the offset between the stationary and non-stationary marginal lines. These climate change effects are also reflected in MCP (Fig. 1c), which shows higher values for earlier years in case of stronger climate change. Comparing the marginal to the conditional cumulative probabilities in the four quadrants of Fig. 1c conveys the strong influence of historical record values. Probability measures aggregated over the entire distribution, like MCP, result in record-breaking probability estimates that greatly differ from history-aware ones that take into account the current record. In case of a high current record, the probability of breaking it in the near term lies far below the marginal average probability (top quadrants Fig. 1c), whereas a low current record subjects a region to much higher-than-marginal probabilities of near term record-breaking events and potentially impacts (bottom quadrants Fig. 1c). In addition, Fig. 1c shows the practical relevance of CCP as a metric of record-breaking probability through time: by assessing record breaking by a certain year $N$, the metric captures both the effects of the historical record–mostly governed by natural variability–as well as the effects of the emerging forced climate change trend.

Figure 1b and c feature a vertical line indicating the year 2050; in the following we show spatial patterns of record-breaking probabilities illustrated by MCP and CCP in 2050.

## Global observed current records

The observed state of Rx1d records, based on the period 1950–2015 for the observational and reanalysis datasets, HadEX3[52], REGEN[53], and ERA5[54] (Fig. 2) govern the conditional probabilities discussed above. HadEX3 and REGEN are based on ground observations and thus have only partial coverage; we include all gridcells that have an Rx1d value for at least 63 out of the 66 years between 1950–2015. For ERA5, which has global coverage, we use all gridcells over land. We evaluate records at the single gridcell level ($1.875° \times 1.25°$ for HadEX3 and $1° \times 1°$ for REGEN and ERA5, i.e., a few 100 km across), since many disasters affect areas of such scales or smaller, and communities and governments act to build resilience on such scales as well[50]. Another reason not to average over larger regions is the high small-scale spatiotemporal variability of Rx1d. For assessment of records in extreme indices that vary on larger spatial scales, such as extreme temperature anomalies, spatial aggregation can be useful, as done by e.g., Fischer et al.[22] and Thompson et al.[34].

Observed Rx1d record levels ($L$, Fig. 2a–c, anomalies w.r.t. 1950–2015) follow a rather smooth and well-understood climatological structure, showing the highest precipitation anomalies in the tropics and monsoon regions, and reflecting locally high orographic precipitation around mountain ranges. The record-setting year $T$ and record quantile level $F_{T,obs}(L)$ in Fig. 2d–f and g–i, respectively, however, show a less interpretable spatial pattern without apparent large-scale structure. This is expected because $L$ is determined largely by climatological features, whereas the record quantile levels are determined relative to the local distribution and thus reflect local variability. In addition, because precipitating systems exhibit strong spatial gradients at relatively small scales, record precipitation rates do not affect areas spanning multiple observed gridcells. This means that records in adjacent gridcells are in most cases set by different events and therefore have different occurrence years and quantile levels, leading to the high spatial variability in the pattern. Nevertheless, large-scale variability may lead to random regional features in the historical record years and quantile levels[37-39]. For example, the position of the North Atlantic storm tracks might have, by chance, spared a particular latitude in Northern Europe throughout the length of the observational timeseries, or a subregion on the east coast of Central America, prone to tropical cyclones[62], might have not experienced landfall in the last 50 years. A complete analysis of the event history leading up to the current state in each region is outside the scope of this study, but we can visually identify regions with low historical record quantile levels (darker colours in Fig. 2g–i) and long record gaps (darker pink in Fig. 2d–f). In HadEX3, Australia features many low-record regions, and also South-America, Central-America Russia, Norway and West-Africa show darker clusters. REGEN – having a higher resolution, different coverage, and different gridding and interpolation procedures[53] – does not agree in all locations, but also features dark clusters in Australia, Central-America, and (different regions of) South-America. The full land coverage of ERA5 shows how strong small-scale variability in the record pattern is, with two high-record regions (light colours) in

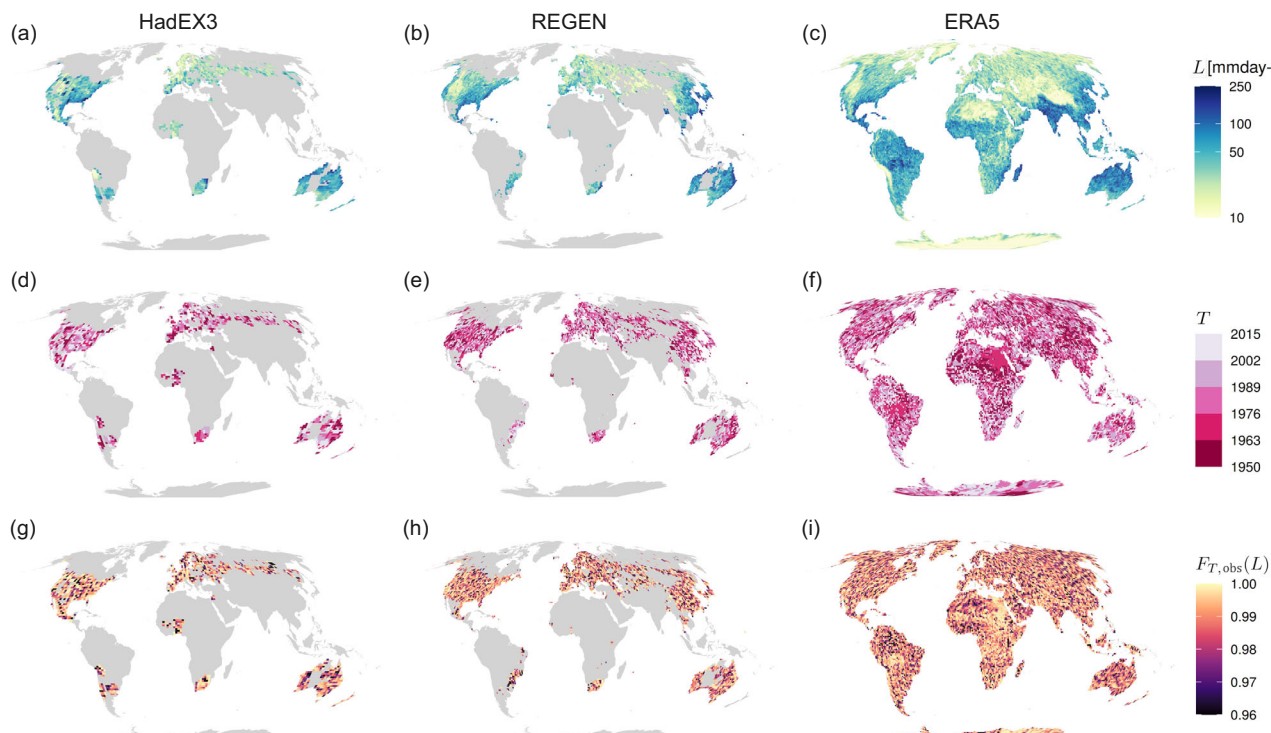

**Fig. 2 | Overview of historical extreme precipitation record metrics for observational and reanalysis data. a–c** Annual maximum daily precipitation (Rx1d) record level $L$ as anomaly relative to 1950–2015 mean Rx1d. **d–f** Year of occurrence $T$ of last Rx1d record in observed period. (**g-i**) Record quantile level $F_{T,\text{obs}}(L)$ of record $L$ in year of occurrence $T$. Figures based on data from HadEX3 (first column), REGEN (second column) and ERA5 (third column), coverage period 1950--2015.

middle South-America and northeastern Africa as its most prominent features. Importantly, all three datasets show that there is no governing spatial or temporal pattern of quantile levels: low records and long record gaps can occur anywhere. See Sect. 4.4 and Supplementary Fig. S2 for an evaluation of the agreement in record statistics between datasets.

We emphasise that the specific quantitative values in Fig. 2g–i) are subject to large uncertainties and should be interpreted with caution. The main reason for uncertainties and noise in the record quantile level pattern is low observational data availability, leading to uncertainties in primarily the tails of the fitted GEV distributions and thus in observed record quantile levels. Supplementary Fig. S5 shows an example of a record quantile level pattern for the ACCESS-ESM1-5 climate model 40-member ensemble[63], where more accurate local GEV distributions can be fitted. The higher data availability for large ensemble climate models results in more interpretable, smoother quantile level patterns. Qualitatively, the pattern of observed Rx1d record quantile levels would look similar to Supplementary Fig. S5 if we were able to determine the true observational GEV distribution. Sect. 4.2, 4.4 and Supplementary Note S2 contain an extensive discussion of the implications of observational GEV biases.

**Both natural variability in historical records and climate change strongly affect near-term record-breaking probability**

The fields shown above and equations (1)–(3) are used to compute the CCP and MCP from 2016–the first unobserved year–to 2100 for each gridcell. $\Pr(Y_{\text{NR}}{\leq}2050|L)$ thus reflects the probability of breaking the Rx1d record set in the period 1950–2015 in any year between 2016 and 2050. Its pattern shows the imprint of both natural variability and climate change in Rx1d (Fig. 3a–c). Shared across HadEX3 and REGEN, West-Africa, northeastern Australia, Scandinavia and (non-overlapping coverage in) South-America show clustering of high CCP values. These high CCP values coincide with locally low historical record quantile

levels (see Fig. 2c, f), and thus high initial record-breaking probabilities, which also quickly increase due to strong climate change effects on extreme precipitation in tropical, monsoon and high latitude regions[16,64,65]. The full coverage of ERA5 (Fig. 3c) shows clearly how local CCP values are amplified or weakened as a result of natural variability in the historical record.

Egypt and Sudan, and the Gran Chaco area in the middle of South-America feature exceptionally low CCP values. For both these regions, the low CCP values can be explained by very high historical record quantile levels, see Fig. 2j. The regions share a semi-arid climate with sporadic excessive downpours, associated with hard-to-fit GEV distributions with very thin, long tails, making quantile level estimation particularly uncertain[59]. However, absolute Rx1d anomaly values corresponding to the record quantile levels are also high, increasing plausibility of the quantile level estimation (Fig. 2g).

An important insight from Fig. 3a–c, is that historical natural variability, governing the local record quantile levels, has a strong influence on local CCP values. Comparing CCP patterns in Fig. 3a–c to unconditional MCP patterns in Supplementary Fig. S7, which disregard the historical record value and show only the climate change effect on record-breaking probability, confirms that natural variability and climate change have effects of similar magnitude on record-breaking probabilities on multi-decadal timescales. For near-term record-breaking, natural variability outweighs the influence of climate change. High CCP values are not confined to regions where extreme precipitation changes are strongest: local CCP can reach very high values virtually anywhere through the combination of natural variability and climate change. This is similar to Thompson et al.'s[34] conclusion for heatwave risk. The strong influence of natural variability on record probabilities and biased risk perception (see next section) should be taken into consideration in adaptation planning.

As mentioned previously, the low data availability for observational GEV distributions leads to uncertainties in quantile levels that

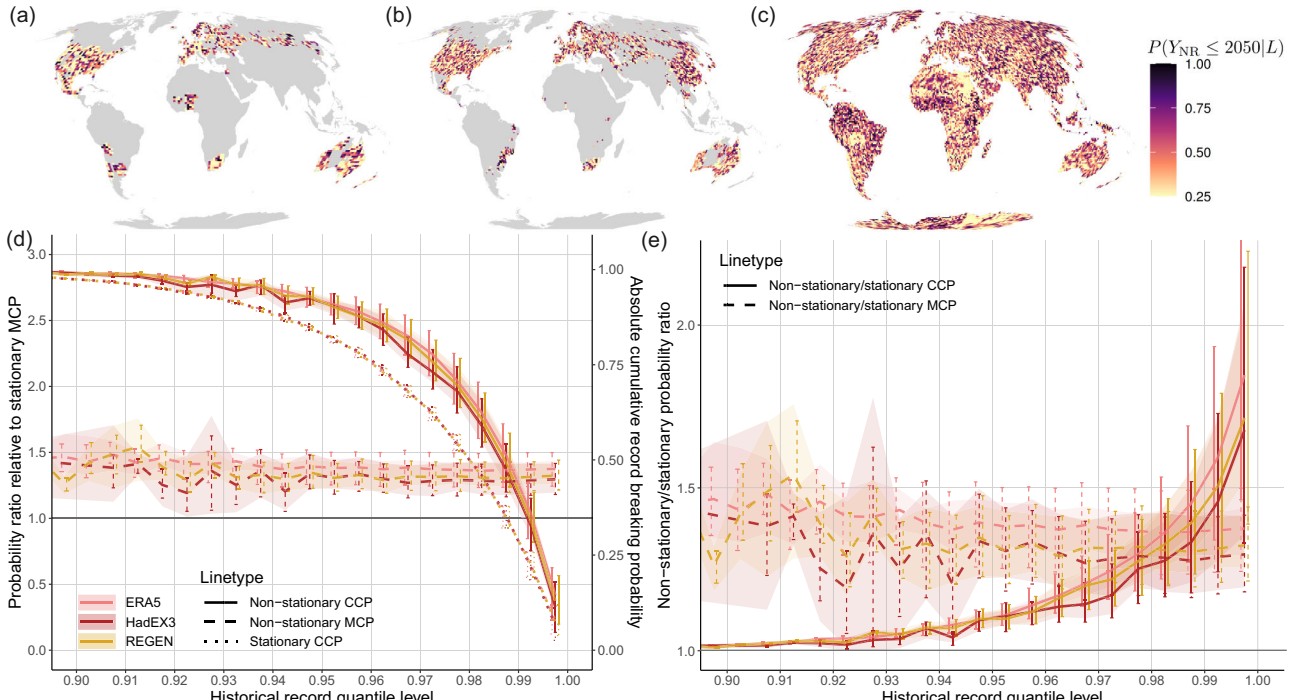

**Fig. 3 | Future cumulative record-breaking precipitation probabilities, binned by observed record quantile level. a–c** Conditional cumulative record-breaking probabilities (CCP) in 2050 conditional on the local record set in the observational period 1950–2015, i.e., the probability that the local observed annual maximum daily precipitation (Rx1d) record is broken in any year between 2016 and 2050, for HadEX3 (**a**), REGEN (**b**) and ERA5 (**c**). **d, e** Contributions of natural variability and climate change to cumulative record-breaking probabilities. **d** Probability ratios relative to the counterfactual stationary marginal cumulative record-breaking probability MCP: non-stationary MCP ratio (dashed) shows mean climate change effect, stationary CCP ratio (dotted) shows effect of natural variability only, non-stationary CCP ratio (solid) shows effect of both natural variability and climate

change (**e**) Probability ratios of non-stationary cumulative record-breaking probability relative to their counterfactual stationary counterpart: MCP ratio shows effect of climate change on marginal record-breaking probabilities, CCP ratio shows effect of climate change on conditional record breaking probabilities. In (**a–c**) the multi-model mean CCP is shown, see Fig. 8 for uncertainty quantifications, in (**d, e**) the line shows the multi-model mean of the bin median and shading represents the spread stemming from across-model differences in the fitted non-stationary generalised extreme value distributions. Error bars show interquartile range of all gridcells in bin in question. Low quantile level bins contain few data-points, resulting in higher variability in the ratio estimates, especially for the lowest resolution dataset HadEX3.

also propagate into CCP patterns. Specific CCP values at individual grid cells can be subject to large errors and should be interpreted with caution, see Fig. 8 for CCP uncertainty quantifications. However, we are confident that our conceptual framework for record-breaking projections has skill based on extensive validation in a model-as-truth setup, see Sect. 4.4. Advancements in observational extreme value distributions, e.g., using higher-accuracy weather model data or artificial intelligence[36,66,67], are needed to increase applicability of our framework to the real world.

In light of the seemingly comparable roles of natural variability and climate change for CCP shown above, we separate their contributions. We determine stationary and non-stationary MCP and CCP as described in Sect. 2.1, and compute probability ratios between these quantities as a function of historical record quantile level. This approach separates effects of natural variability and climate change on record-breaking probabilities, since there is no spatial correlation between the pattern of record quantile levels (natural variability) and that of mean climate change.

The strong effect of the historical record quantile level on record-breaking probabilities in general and on their sensitivity to climate change is evident in Fig. 3d–e, showing probability ratios for 2050 as a function of historical record quantile level. The median of all gridcells within the respective record quantile level bin is shown, and the interquartile range is indicated by the error bars.

In Fig. 3d all probability ratios are determined relative to the counterfactual stationary MCP. The ratio between non-stationary MCP and counterfactual stationary MCP represents the effect of climate

change on record-breaking aggregated over all historical record quantile levels; The ratio between stationary CCP and stationary MCP constitutes the effect of natural variability alone; and the ratio between non-stationary CCP and stationary MCP constitutes the combined effect of natural variability and climate change on the total, local cumulative record-breaking probability.

The stationary MCP in 2050 is 0.34 (see Sect. 2.1 and equation (4)). Note that stationary MCP is independent of climate change and record quantile, and therefore the same for every gridcell. The dashed lines show that the non-stationary MCP is $\approx 1.4$ times larger than the stationary MCP, indicating that the mean chance of breaking an Rx1d record by 2050 has increased by 40% due to climate change. The MCP probability ratio is, by construction, independent of record quantile, as evidenced by 3d and e. The variability in probability ratios for quantile levels below $\approx 0.94$ is due to the low number of grid cells with such low records. The stationary CCP probability ratios (dotted) indicate that natural variability alone leads to a CCP higher than the stationary reference MCP (ratio $> 1$) in regions with record quantile level $\leq 0.987$. This number is no coincidence: the average observed record quantile level is $\approx 0.987$, corresponding to the expected value of $1 - \frac{1}{66+1} = 0.985$ given the observational timeseries length of 66 years (see Supplementary Note S4 for a derivation).

The non-stationary CCP probability ratios (solid) are considerably larger than their stationary counterpart (dotted), showing that climate change affects the Rx1d distribution in a way that rapidly increases the likelihood of event intensities that set records in the current climate. More importantly, about 45% of observed gridcells feature record

quantile levels where non-stationary CCP ratios exceed non-stationary MCP ratios (dashed) (see supplementary Fig. S6). These locations will thus be exposed to higher actual record-breaking probabilities than a marginal non-stationary record-breaking metric that ignores current records would suggest. On the other hand, for the other 55% of grid cells with higher record quantile levels, actual record-breaking probabilities are smaller than the marginal non-stationary MCP estimates. The large fraction of gridcells for which conditioning on the local record leads to larger record-breaking probabilities than the marginal value emphasises the necessity to include local historical observations for accurate hazard estimation and risk assessment.

The non-linear quantile-dependent effect of climate change on CCP manifests clearly in probability ratios of non-stationary CCP relative to stationary CCP, shown in Fig. 3e. This ratio (solid lines) indicates how many times more likely record-breaking in the 2016–2050 time window becomes due to climate change as a function of historical record quantile level. For comparison, we also show the quantile-independent effect of climate change in non-stationary/stationary MCP ratios (dashed, same as in Fig. 3d).

The effect of climate change is minor for low record quantile levels, as expected, since these lead to CCP values of almost 1 by 2050, even in the stationary case: if the current record happens to be very low, it will likely be broken by 2050, even without climate change. As historical record quantile levels increase, the effect of climate change becomes progressively stronger: the rarest extremes become more likely at the highest rate.

About 75% of gridcells have record quantile levels $> 0.9825$ (see Supplementary Fig. S6), for which the non-stationary/stationary CCP ratio exceeds the non-stationary/stationary MCP ratio, meaning that quantile-dependent climate change effects exceed the quantile-aggregated climate change effect on record-breaking probabilities. For the highest historical record quantile levels, climate change makes record-breaking by 2050 about 75% more likely compared to a counterfactual stationary climate. This conditional probability increase is almost twice as large as the marginal probability increase that disregards historical record levels (dashed). The exact relationship between climate change and CCP is convoluted since both the Rx1d increase as well as the cumulative record-breaking probability are non-linear functions of the record quantile, yet the effect of climate change on CCP is robust across the observational and reanalysis datasets. Figure 3e shows the median ratio over all gridcells with a given historical record quantile; as the magnitude of climate change varies locally, this ratio can be higher in regions with strong climate change and lower in regions with weak climate change, as shown by the error bars indicating the interquartile range. The stronger sensitivity of the most extreme events to climate change, found in several studies prior to this one[12,32,33], has important implications for impact assessment, e.g., in the context of so-called record-shattering events[22–24]. The above CCP values are dependent on the choice of 'end year' (2050 in this case) for the cumulative summation, however, results are qualitatively robust.

The finding that climate change effects on record-breaking probability become progressively stronger with record quantile level is important: regions with high current record quantile levels are, while having low absolute record-breaking probabilities, subject to the strongest relative increases in CCP. In other words, regions that might consider themselves safe because of a low current record-breaking probability become less safe at the highest rate.

### Disaster potential as a combination of high record-breaking probability and low risk awareness

The sections above showed that conditioning on local historical records changes natural hazard probability quantification, and hence downstream risk assessment. As mentioned in the introduction, disaster risk is also sensitive to effects of the historical record evolution

on risk perception, and hence on disaster preparedness and resilience. Regions where the last observed Rx1d record was low and a long time ago might be considered lucky not to have experienced precipitation related disasters in recent history. This luck, however, also implies that such regions are at present exposed to high record-breaking probabilities, for which they are possibly not prepared due to lack of experience, biased risk perception and resulting insufficient resilience building[43–49]. For a more comprehensive view on regions at risk for precipitation caused disaster, we extend the above analysis with a qualitative assessment of disaster preparedness based on the historical evolution of precipitation records.

We define the metric 'state likelihood' as the probability of not having broken the last observed Rx1d record between its year of occurrence $T$ and the last observational timestep 2015, i.e., the probability of the true historical trajectory. A low state likelihood indicates that the historical precipitation timeseries features unusually low and/or few recent records. The lower the state likelihood, the larger the possible discrepancy between local extreme precipitation-related risks and awareness of and preparedness for those risks.

To compute the state likelihood, we perform the CCP calculation outlined above in equations (1)–(3), but determine the cumulative sum from $T + 1$ to 2015, instead of 2016–2100. This gives the gridcell-specific probability of breaking the current record in any year between the record-setting year $T$ and the last observed year 2015. The likelihood of the true state of not having broken the record is 1 minus this probability, see equation (5) and Sect. 4.3 for more details.

$$\text{State likelihood} = 1 - \Pr(T < Y_{NR} \le 2015 | L) = 1 - \sum_{n = T+1}^{2015} \Pr(Y_{NR} = n | L)$$

(5)

State likelihood thus constitutes a measure of disaster potential by combining the quantitative likelihood that a record-breaking precipitation event occurs, governed by the historical record quantile level and climate change, with the qualitative likelihood that it causes a disaster, influenced via societal processes of disaster memory, experience, and (biased) risk perception.

In HadEX3 (Fig. 4a) notable regions with clusters of low state likelihood include north- and southeastern Australia – including the major cities of Brisbane and Sydney –, West-Africa–including the metropolitan corridor between Abidjan in Ivory Coast and Lagos in Nigeria[68] –, and southern France. In REGEN (Fig. 4b) we also see low state likelihood at those locations, and the higher coverage of REGEN in Asia shows low state likelihood in Inner Mongolia/Northern China as well. In the ERA5 (Fig. 4c) pattern clusters are less clear, but hints of clustering in Asia and Australia—corresponding to HAdEX3 and REGEN–and also Brazil can be distinguished. All datasets show ample scattered gridcells with low state likelihood throughout their coverage, with noisiness that is partly due to the small observational sample sizes mentioned in previous sections. Also here, large uncertainty margins necessitate caution when interpreting specific local values.

In Fig. 4d–f, gridcells where low state likelihood ($\le 0.5$) and high CCP by 2050 ($\ge 0.75$, see Fig. 3) co-occur are highlighted. Where both hold (green), a low current record, climate change, and potentially biased risk perception combined lead to high near-term disaster potential. It is possible that local communities in these gridcells are unprepared for the record precipitation events that are likely to happen in the near future. The ERA5 coverage shows most clearly that the high CCP gridcells (yellow) are frequently located around the tropics, where extreme precipitation increases strongly with climate change[65]. The pattern of low state likelihood (blue) and thus the co-occurrence pattern (green) is patchier in the tropics, most likely due to the high variability of tropical Rx1d. In eastern Asia, there is widespread co-occurrence, which is in line with the robust, high signal-to-noise ratio change in Rx1d in this part of the world[20]. Overall, high disaster

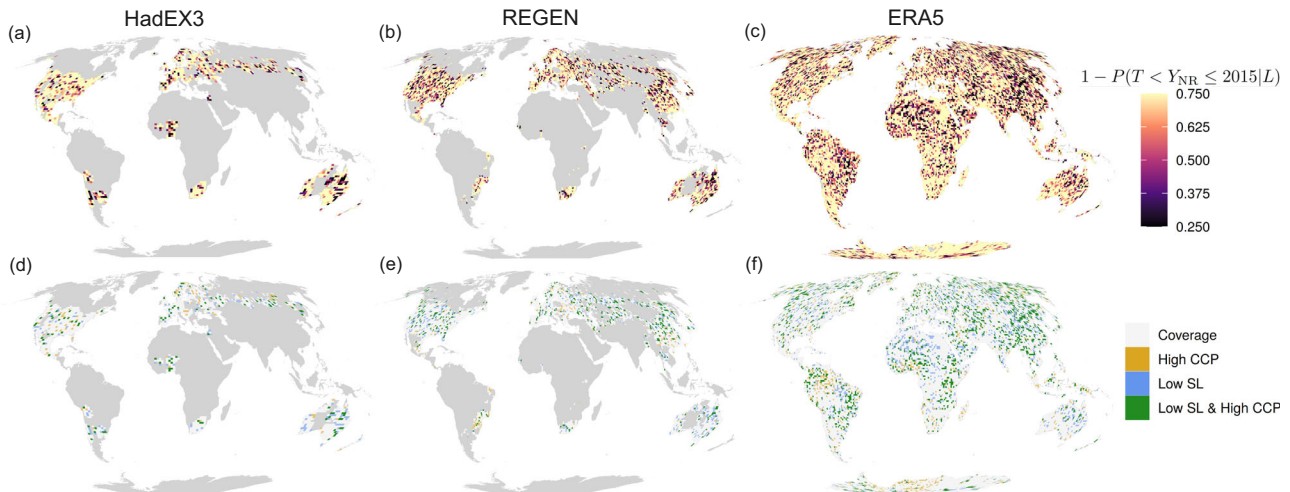

**Fig. 4 | Observed state likelihood and disaster potential. a–c** 2015 state likelihood, defined as the likelihood of the true real-world state in which the precipitation record has not been broken between the last observed record in year $T$ and the last timestep of observational/reanalysis data (2015). Note the colour scale is inverted relative to Fig. 3 to maintain the convention that darker colours mean

higher risk. **d–f** High disaster potential gridcells with 2050 cumulative conditional record-breaking probability (CCP) ≥0.75 and state likelihood (SL) ≤0.5. Figures based on data from HadEX3 (first column), REGEN (second column) and ERA5 (third column) and eight CMIP6 models forced with SSP2-4.5.

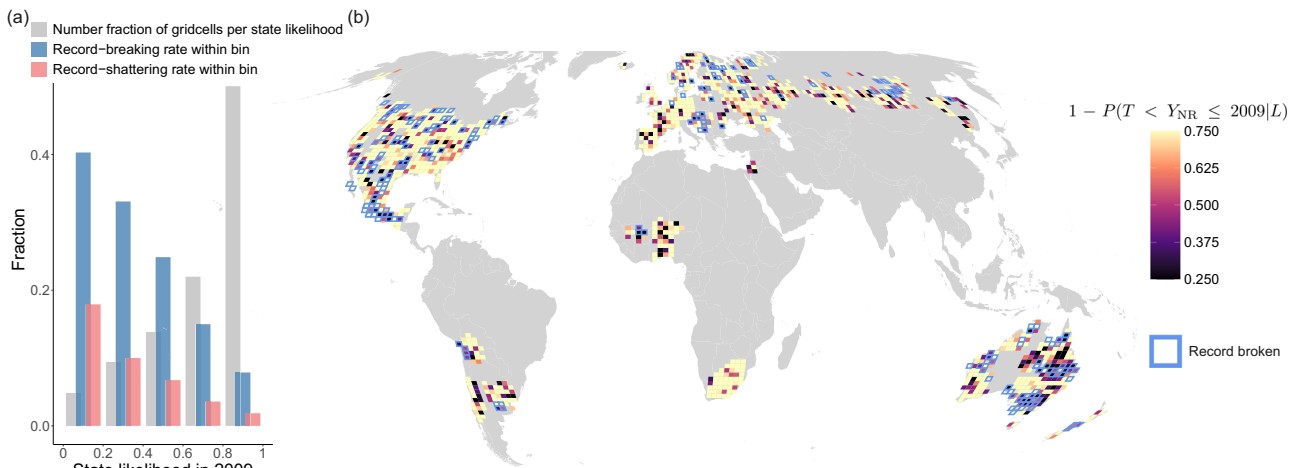

**Fig. 5 | Evaluation of the association between state likelihood and disaster potential. a** Fraction of gridcells within 2009 state likelihood bin (*x* axis) in grey, fraction thereof that breaks (shatters by one standard deviation) their record in 2010–2015 in blue (pink). **b** 2009 state likelihood pattern

with gridcells that break their annual maximum daily precipitation (Rx1d) record in 2010–2015 highlighted in blue. The areas in the grey boxes are referred to below in exemplary event case studies.

potential regions (green) are are scattered around the globe, but form clusters in eastern Asia, Brazil, and Australia. These areas contain fast growing subregions that are particularly vulnerable to impacts of natural disasters[47].

For validation of the relevance of state likelihood for disaster potential, we conduct the state likelihood computation for HadEX3 for the period 1950–2009, and assess the relationship between 2009 state likelihood and records that occur in the following final years of the timeseries 2010–2015. The number of gridcells exhibiting a certain state likelihood decreases for lower state likelihoods (Fig. 5a, grey bars), but the relative frequency of record-breaking in 2010–2015 among those gridcells increases (blue bars). The latter is a result of the association of low state likelihood with low historical record quantile level and thus high record-breaking probabilities. In addition, the rate of record-shattering (breaking a record with a margin of at least one standard deviation, pink bars), albeit low, increases as well with lower state likelihood; a low current record is more likely to be shattered by a

large margin, further increasing the potential for excessive damage. Also, low state likelihood in models is correlated with earlier occurrence of the next record event and a higher numbers of new record events between 2016 and 2100 (Supplementary Fig. S8), further verifying the quantitative component.

We verify the qualitative relationship between state likelihood and disaster preparedness using real-world disaster information for 2010–2015. The 2009 state likelihood (shading) is shown in Fig. 5b, with gridcells that break the record in 2010–2015 highlighted by blue frames. After evaluation of the years in which record-breaking occurs in HadEX3, we selected the clusters in the grey boxes, which contain ≥2 gridcells that have 2009 state likelihood ≤0.25, and exhibit record-breaking in the same year.

There are two clusters with record-breaking in 2010/11 in Australia–one in southeastern Queensland and one in Victoria, one with record-breaking in 2010 in Central America, and one involving New York City and New England with a 2011 record-breaking event. A

simple internet search using the search term '[region], [year], flood' yields reports of distinct events with disastrous consequences that plausibly correspond to the record-breaking events recorded in HadEX3[69–72]. HadEX3 precipitation records cannot be traced back to single meteorological events, however, since the gridding scheme used to create Rx1d data breaks the spatiotemporal coherence[52]: station data from different days may be aggregated in the Rx1d value in one gridcell. To ensure the relationship between the record-breaking in HadEX3 and the reported disasters, we evaluated daily precipitation station data in the gridcells of interest and found clear peaks on the exact days that correspond to the events that the internet search yielded, see Supplementary Fig. S9. Also the readily available flood reports for record-breaking year-region combinations from HadEX3, and absence thereof for random year-region combinations, provide evidence that the record-breaking events in HadEX3 were associated with the reported disasters.

In austral summer 2010–11, southeastern Queensland, including the third most populated Australian city of Brisbane, was impacted by sustained precipitation leading to floods affecting 2.5 million people and incurring over 5 billion AUD of damages. In the same summer, also Victoria experienced record rainfall, widespread flooding, and severe damages[70]. A commission tasked with evaluating the government's response to the Queensland event stated that "the disastrous floods which struck south-east Queensland in the week of 10 January 2011 were unprecedented, in many places completely unexpected, and struck at so many points at once that no government could be expected to have the capacity to respond seamlessly and immediately everywhere, and in all ways needed." [69]. Based on probability estimates, however, it was likely that precipitation records would be broken by potentially large margins in this region. This quote thus exemplifies how risk awareness and preparedness ("completely unexpected") can diverge from actual risk, and thereby result in damage that might have been prevented. Although associated Rx1d records are found in HadEX3, it is important to note that both Australian events were pre-conditioned by persistent rain and saturated soils, likely exacerbating the damage.

The 2011 record-breaking precipitation in New Jersey and New York City was associated with Hurricane Irene, which was monitored extensively while making its way northward to North Carolina and New England. Its path and associated weather was thus well-forecast, but the severity of the storm exceeded the disaster response capacity. Especially in southern Vermont (top right of grey box, Supplementary Fig. S9) enormous infrastructure damage resulted, leading to claims that Irene's impacts rank second after the 1927 floods, the greatest natural disaster in Vermont, which suggests once again that the long record gap might have negatively affected preparedness[72].

The 2010 event in Central America, associated with tropical storm Agatha which made landfall on the western Mexican-Guatemalan border, was well-forecast[71]. Nonetheless, the rarity of tropical cyclones making landfall on low-latitude western coasts explains the lack of recent experience with such events and was associated with relative unpreparedness of society and major damages[62,71].

Of course, there is no way of knowing if the same weather events would have been less disastrous if the affected regions had experienced more recent and/or more severe record-breaking events. We also find damage reports for clusters of record-breaking events in regions without low state likelihood, such as hurricane Odile in Baja California in 2014 and the Warum floods in western Australia in March 2011[73,74]. These floods were not as disastrous as the ones we addressed above, but we cannot attribute this to state likelihood given the low number of events and our simple anecdotal verification method. Thus, we emphasise that our framework relating state likelihood and disaster preparedness is an interpretive, conceptual framework and does not provide a quantitative, empirical threshold beyond which risks measurably increase. However, we consider the statistical and anecdotal evidence compelling for an association between low state likelihood and the potential for precipitation-caused disasters, through the combination of high record-breaking probability and potentially low risk awareness and preparedness. We hope to offer food for thought towards more comprehensive approaches to climate risks and disaster resilience that integrate historical experience, current and future climate information, and socio-political and economic factors.

## People exposed

Lastly, we translate the rather abstract probabilities shown to the more interpretable metric of people exposed to high disaster potential. Gridded population numbers for SSP2 are used, provided by the Socioeconomic Data and Applications Centre (SEDAC)[75,76]. We do not show HadEX3 and REGEN as their sparse coverage gives an incomplete picture; in ERA5 all land areas and therefore also all people are covered. Part of the regions where the effects of climate change on extreme precipitation are particularly strong, such as South-East Asia and West-Africa, have experienced fast population growth, economic growth, and urbanisation in the last few decades, leading to the concentration of large numbers of people and dense infrastructure[77]. Such fast development is associated with adaptation gaps and absence of cross-generational disaster memory, making these growing populations vulnerable to precipitation induced disaster[46–48,78].

The number of people exposed to a non-stationary CCP ≥0.75 starts to rise from about 2030 onwards, reaches half a billion by 2060, and over one billion by 2100 (Fig. 6a). As in Fig. 3 we show non-stationary and stationary MCP and CCP to visualise the effect of natural variability and climate change and their interaction. Also in a counterfactual stationary climate (dotted), natural variability leads to exposure of people to high CCP values over time, however, the rate of increase is much lower.

Comparing non-stationary CCP ≥0.75 exposure (solid) to non-stationary MCP ≥0.75 exposure (dashed) reveals an important insight. For MCP (not conditioned on the historical record), the number of exposed people starts to increase 20 years later than for CCP. This shows that probability estimates based on mean climate change effects greatly underestimate near-term global exposure to record-breaking precipitation risk, since natural variability in the local Rx1d observations is responsible for much of the near-term high record-breaking probabilities. This is due to regions with low local current records reaching high CCP values very quickly.

Interestingly, the MCP increase is steeper than for CCP. Conditioning on the historical record quantile level damps the rate of exposure increase, since CCP ≥0.75 regions emerge gradually as a function of their historical record quantile level. For MCP, emergence is more binary: the mean climate change pattern results in large regions attaining MCP values ≥0.75 at once when a certain global warming level is reached (see Supplementary Fig. S7).

A noteworthy feature of Fig. 6a is the smaller model spread (coloured shading) for CCP compared to MCP. Again, this is due to the large dependence of the CCP evolution on record quantile level. The first regions to attain CCP values ≥0.75 have low historical record quantile levels causing steep near-term increase in CCP (see Fig. 1), even without strong climate change. Hence, for early years, model spread in climate change is not reflected in the emergence of exposure to CCP ≥0.75. As time progresses, regions with lower historical record quantile levels start to emerge, where climate change contributes to greater degree to increases in CCP. Therefore, the effect of model spread on CCP ≥0.75 exposure increases with time. Emergence of high MCP exposure is dependent on climate change only, and therefore more sensitive to the model spread therein, reflected by a larger uncertainty range at all times. The combined uncertainty range resulting from uncertainties in ERA5 GEV fits and model differences (light grey shading) is substantial, but does not change the main

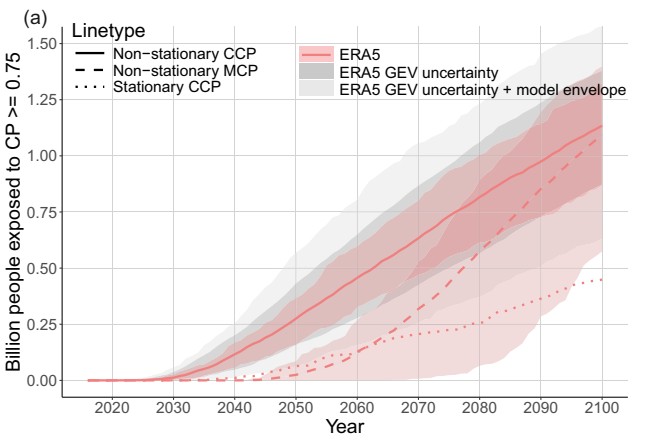

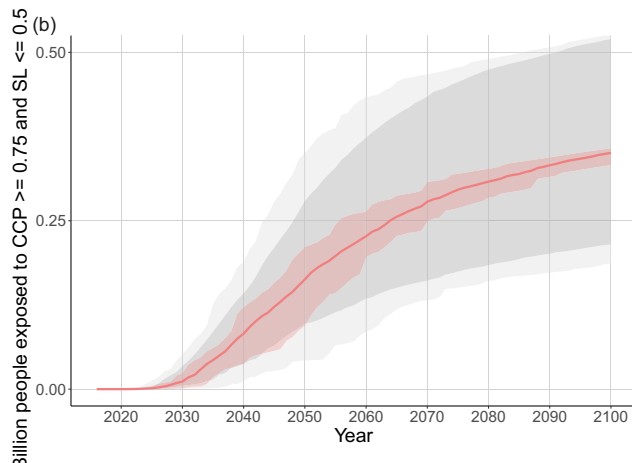

**Fig. 6 | People exposed to high disaster potential globally for future years.** **a** Number of people living in regions with a marginal cumulative record-breaking probability (MCP) or conditional cumulative record-breaking probability (CCP) of 0.75 or higher as a function of year $N$ (see equation (3)) for ERA5. Non-stationary and stationary CCP and MCP are shown, reflecting the impact of climate change on marginal and conditional probabilities. **b** People exposed to high disaster potential, reflected by regions where state likelihood ≤0.5 and CCP ≥0.75, see Fig. 4f. In both subplots, lines show multi-model mean, coloured shading shows model mean envelope, dark grey shading shows multi-model mean CCP uncertainty due to ERA5 generalised extreme value distribution (GEV) fit uncertainties and light grey shows combined ERA5 GEV uncertainty and model envelope (see Fig. 1 caption and Sect. 4.2).

finding that a considerable fraction of the world population will be exposed to high conditional record-breaking probabilities.

Finally, Fig. 6b shows the time evolution of people exposed to high precipitation disaster potential, defined as 2015 state likelihood ≤0.5 and CCP ≥0.75. In other words, the number of people living in green gridcells in Fig. 4f as a function of time. As in Fig. 6a, there is a steady increase starting around 2030. The exposure curve flattens as all the gridcells with low 2015 state likelihood attain CCP values ≥0.75, and only population growth contributes to increasing exposure. In 2100, a worryingly high number of 0.3 billion people lives in an area where high record-breaking precipitation probabilities and low disaster preparedness coincide. Combined observational GEV uncertainties and model spread suggest a number between 0.2 and 0.5 billion people.

## Discussion

In this study we evaluate regional future record-breaking precipitation probabilities conditional on historical precipitation observations, using HadEX3 and REGEN gridded observations, ERA5 reanalysis, and CMIP6 climate models forced with a moderate warming scenario. We estimate the probability of breaking the local observed Rx1d record by some future year $N$. This metric combines the historical record level and local projected changes to the Rx1d distribution under climate change. We use non-stationary extreme value theory to determine observed record quantile levels and climate model based future evolution of the extreme precipitation distribution. This study is distinctive in its focus on local record-breaking precipitation probabilities conditional on the observed historical record level, rather than marginal record-breaking probabilities independent of observations.

High spatiotemporal natural variability in Rx1d (annual maximum daily precipitation) leads to a large range of historical record levels of varying rareness. Regions with low current record quantile levels and strong climate change are exposed to the highest current and future record-breaking probabilities. Low record quantile levels can occur anywhere, and the historical evolution of precipitation leads to clusters of low record quantile levels and associated high record-breaking probabilities in, among other regions, West-Africa, (north)eastern Australia, eastern Asia, and Brazil. Climate change contributes to increasing record-breaking probabilities in these regions, particularly strongly around the intertropical convergence zone.

Natural variability in historical precipitation observations strongly affects local future record-breaking probabilities, and probabilities conditioned on historical records greatly differ from those based on patterns of future climate change trends. Importantly, regions with high current record quantile levels show relatively small future record-breaking probabilities, and thus might consider themselves safe. Yet, climate change affects the precipitation distribution non-linearly, leading to a stronger relative increase in record-breaking probabilities in those regions where the current record is high. In other words, these regions become less safe at the highest rate. Nonetheless, regions with very low historical record quantile levels feature the highest absolute record-breaking probabilities. Because of their strong influence on record-breaking event probability, it is key to include historical observations for actionable local climate information.

Over one billion people globally would be exposed to higher than 75% probability of experiencing a local Rx1d record-breaking event by 2100, according to SSP2-4.5 population growth and climate change. Based on the patterns we find, a large fraction of the exposed people lives in developing economies, featuring strong population and economic growth, associated with rapid urbanisation and adaptation gaps[47,78]. In combination with fewer observations, higher model uncertainties, and less climate research being executed in these regions, this is a major challenge for climate justice.

The observed precipitation history also affects local risk perception. We introduce a qualitative measure of disaster preparedness based on societal processes affecting disaster resilience through risk perception biases and (lack of) historical experience[42,43,45–48]. Regions where historically observed record levels are low and record-breaking events have not happened in recent history are more likely to have low risk awareness and lack of preparedness, exposing these regions to a higher risk of precipitation caused disaster. (North)Eastern Australia, eastern Asia, and northern South America might be subject to particularly high disaster potential given the combination of high near-term record-breaking probabilities and low disaster preparedness due to disaster memory effects, as well as smaller scattered clusters in the Northern midlatitudes.

All the results we show are affected by the difficulty associated with the estimation of GEV distributions based on the small observational sample sizes available, but we are confident that our results are

qualitatively robust based on extensive validation using climate models as well as alternative sources of disaster information.

In a wider context, our results are relevant to disaster governance; the process of continuously assessing, mitigating and adapting to risks, aimed at preparedness and resilience for future disasters. At present, it is still all too common to increase risk mitigation and adaptation efforts only after the occurrence of disaster, and only temporarily. Our results suggest that a growing fraction of the world population will live in regions with high near-term probability of record-breaking extreme precipitation while likely lacking adequate disaster governance practices. We emphasise the need for increased efforts to integrate transdisciplinary and transsectoral disaster governance practices in global and local policies to increase resilience to the threat of unprecedented extreme precipitation, especially in vulnerable regions.

## Methods

### Data

**Observational and reanalysis data.** The variable we focus on in this study is annual daily maximum precipitation (Rx1d). Floods are often preconditioned by e.g., sustained rain for multiple days, saturated soils due to wet conditions over weeks to months preceding the flood, or snow melt, yet, in all these conditions high short-duration (daily or even shorter) sums of precipitation are either implied or required to create the surge in runoff that eventually triggers flooding[79]. Hence, Rx1d is generally relevant for pluvial flooding.

We use gridded in situ observations from HadEX3[52] at 1.875° × 1.25° resolution and REGEN[53] at 1° × 1° resolution, both running from 1950–2015. Both datasets provide gridded indices, including extreme indices, at annual resolution. We use the masked version of REGEN, where gridcells with unreliable data are masked out. Only those gridcells with a complete timeseries from 1950–2015 are used, given the path-dependence of record-breaking. We use gridded observational products rather than raw station data to increase generalisability across observational, reanalysis and simulated data. Although station data represents the true precipitation (modulo measurement errors) at one single point location more accurately, gridded data enables a more meaningful spatial analysis. The gridding procedure used differs between HadEX3 and REGEN, however; whereas HadEX3 interpolates station Rx1d values, REGEN assimilates data to produce gridded daily records, after which Rx1d is extracted[52,53]. The resolutions of both datasets differ as well. Therefore, we do not expect very high quantitative agreement between both datasets. Our analysis is not dependent on the accuracy of absolute values in physical units as record-breaking is a relative measure, so systematic biases or offsets between the absolute values of the gridded products and the truth are not critical as long as they are consistent within the dataset.

The observational datasets have partial coverage since only gridcells with enough stations in their vicinity are included. We therefore include the reanalysis dataset ERA5[54], which has full global coverage. The data assimilation and modelling procedure used for ERA5 introduces differences with respect to real observations, meaning the agreement between observations and reanalysis is also not expected to be very high. It has been shown that ERA5 has difficulties reproducing observed Rx1d magnitudes, and that the influence of geographical features such as coastlines and orography is not always captured. In addition, the uncertainties in the tropics are high due to low observational availability and model uncertainties[80]. Also desert regions are sensitive to uncertainties and errors, especially in fitted GEVs, due to the large fraction of zeros in their Rx1d samples. Systematic underestimation of Rx1d by ERA5 does not have to be critical for the analysis of record-breaking since records are evaluated within one climate realisation. The relative characteristics (mean vs. variability in tropics and extratropics) within ERA5 are comparable to those in observations[80], hence we expect ERA5 to add useful insights. The

inaccuracies around geographical features need to be kept in mind when interpreting the results. For ERA5, conservatively regrid daily fields with an original resolution of 0.5° × 0.5° onto the 1° × 1° REGEN grid to make the data more compatible with the models used for future projections of record-breaking (see below), and extract Rx1d using Climate Data Operators (CDO[81]). Although ERA5 covers the period 1950–2023, we use data until 2015 to enable comparison to the observational datasets.

For later operations, it is practical to work with local Rx1d anomalies. We centre every local Rx1d timeseries by subtracting its 1950–2015 mean. Historical record (anomaly) levels and record years are determined by extracting the maximum Rx1d value and its corresponding year.

The observational and reanalysis data sets start in 1950 only, and it is imaginable that a value higher than the record since 1950 occurred in years prior to 1950. This leads to a potential overestimation of record-breaking probabilities in our results. At the same time, it is reasonable to start counting records in 1950 given that records older than 75 years are unlikely to still have much influence on infrastructure and response planning or public risk awareness[45].

**Climate model simulation data.** We use eight coupled climate models from Phase 6 of the Coupled Model Intercomparison Project (CMIP6), listed in Supplementary Table S1, for the estimation of future Rx1d distributions and for method validation. Simulated daily precipitation data is conservatively regridded onto the HadEX3 and REGEN grids to enable future projections on the native observational grids, after which annual Rx1d values are extracted and 1950–2015 means are subtracted per gridcell, as done for the observational data. We use the shared socio-economic pathway (SSP) 2-4.5 future scenario since this is most aligned with the current trajectory of society and emissions. Given our aim to provide practically relevant probability estimates based on observed records, the future scenario needs to be plausible as well to maintain real-world applicability of the results.

The eight climate models used are selected based on having at least five ensemble members with historical and SSP2-4.5 forcing and at least 1.25° × 1.875° resolution–the same as HadEX3. These were deemed the minimum requirements for robust gridcell-specific GEV fits (251 years × 5 members = 1255 datapoints per gridcell) as well as compatibility with observational grids. For model ensembles with more than five historical+SSP2-4.5 members we use all available members. For projections on the REGEN grid (used for both REGEN and ERA5), most simulated fields had to be conservatively regridded to a higher resolution, introducing biases. For this reason we mostly focus on model results on the HadEX3 grid. We use the full 1850–2100 historical+SSP2-4.5 period for GEV fitting, but determine record metrics.

**Population data.** We use spatial population data for SSP2, provided by the Socioeconomic Data and Applications Centre (SEDAC)[75,76]. These exist at 10-year intervals. We use the 2010, 2020, 2050, and 2100 data, and interpolate linearly between 2010 and 2020 to obtain the 2015 population density, and again between 2015–2050 and 2050–2100 to obtain population density per year from 2015–2100. We multiply population density with gridcell area obtained from the original files to obtain absolute population counts.

### Generalised extreme value distributions

Generalised extreme value (GEV) distributions are the basis of our probability calculations. GEV distributions describe distributions of block maxima, such as Rx1d (annual daily precipitation maximum), and are defined by three parameters: the location parameter $\mu$, scale parameter $\sigma$, and shape parameter $\xi$ represent centre, spread, and tail characteristics. We use the R-package extRemes[82] to fit non-stationary GEV distributions.

Non-stationarity implies that we let the GEV distribution vary in a prescribed way. In our setup, $\mu$ and $\sigma$ vary as a linear function of covariates $g_\mu(t)$ and $g_\sigma(t)$, which are functions of time and reflect the effects of climate change on the Rx1d distribution. $\xi$ is kept constant with time, since estimation of $\xi$ is associated with high uncertainty[57], and it is common to assume that the tail characteristics of local extreme precipitation distributions remain reasonably constant with (moderate) climate change. The covariates are dependent on the data to which the GEV is fit, and are discussed below. This leads to the following definition of the GEV parameters:

$$\begin{cases} \mu(t) = \mu_0 + \mu_1 g_\mu(t) \\ \sigma(t) = \sigma_0 + \sigma_1 g_\sigma(t) \\ \xi = \xi_0 \end{cases} \tag{6}$$

The five coefficients $\mu_0$, $\mu_1$, $\sigma_0$, $\sigma_1$, and $\xi_0$ are determined using maximum likelihood estimation within the *fevd()*-function of the extRemes R-package[82]. This GEV setup is a commonly used to statistically describe extremes in a changing climate[83].

It is worth noting that Rx1d is only moderately extreme, and therefore not max-stable, meaning that the values do not always fit nicely in one and the same distribution[57]. Yet, GEV distributions are an accepted and useful statistical tool if caution is taken when interpreting the results. We provide extensive contextualisation and validation of our results using non-parametric and empirical analyses, and are confident that the use of GEV distributions, if caution is taken, is justified. In this study, it is impossible to focus on longer block maxima (closer to max-stability) since this would make the observational sample size prohibitively small.

Inclusion of the record event in the GEV fit influences the results: as it is the most extreme event of the timeseries, it is located in the tail and will therefore affect the tail properties of the fitted distribution. It has been shown that excluding the most extreme event from a GEV fit results in overestimation of the quantile level of the event in question (i.e., the event is deemed less likely than it actually is), and including the extreme event leads to the opposite[39,84]. Both effects are undesirable. Whether or not to include the extreme event of interest in the sample used for fitting a GEV is an unresolved question. Most studies on extreme event attribution, e.g., by World Weather Attribution, exclude the event that is being analysed from the GEV fit[85], however, these studies are often motivated by the extremeness of the event, and focus on return periods of very extreme 'almost impossible' events. thus leading to selection bias concerns and very high sensitivity to inclusion of the event[84,85]. In our case, the record of the observed timeseries is not necessarily very extreme, and there are no selection bias concerns since we assess the record in every gridcell, regardless of its extremeness. As the main concerns regarding inclusion of events of interest in the samples for fitting GEV distributions are not critical in our case, while information is added by including the records, we prefer to include them.

**GEV distributions for observational and reanalysis data.** GEV distributions are fitted to observational and reanalysis data using the general scheme described above. Climate change effects on extreme precipitation have been detected in historical observations[2,13,17–20], and we account for this by using global mean surface temperature (GMST,[86]), smoothed with a loess-filter, as the covariate for both $\mu$ ($g_\mu(t)$) and $\sigma$ ($g_\sigma(t)$). Despite the nonlinear relationship of extreme precipitation with temperature, the relatively small warming trend in the historical period and the high degree of variability in the observations justify the use of a simple linear relationship with GMST. The non-stationary covariates are the same for every gridcell.

The robustness of the observational/reanalysis GEV fits is impaired by the short observational timeseries. The largest uncertainty and error in the observational GEV fits is associated with the shape

parameter $\xi$ that governs tail properties. We tested potential GEV fitting improvements based on spatial pooling, i.e., combining data from multiple locations to increase the size of the sample on which the GEV fit is based, to improve the tail representation of the GEV fit so that the observed record quantile levels can be better estimated.

We tested two different variants of spatial pooling; 'shape-only' and 'naive'. In Supplementary Note S2 we provide a detailed assessment of the effects of potential GEV fitting improvements on the GEV parameters and on record quantile levels, but for the sake of space, we only report the outcome here. The preferred method for our purposes is a simple naive spatial pooling scheme: all Rx1d values from a $n \times n$ window around the gridcell of interest (middle) are pooled into one sample which is used to fit a GEV with non-stationary $\mu$ and $\sigma$ and stationary $\xi$ as described above. This GEV is assigned to the middle gridcell, and using moving windows we apply this to every gridcell. The pooling windows we use are $3 \times 3$ gridcells for HadEX3 ($\approx 4.9° \times 3.75°$ deg) and $5 \times 5$ gridcells for REGEN and ERA5 ($5° \times 5°$). In order to prevent pooling of data across sharp climatic regime boundaries (e.g., mountain ranges or coastal gradients), we exclude gridcells where the standard deviation of the Rx1d sample differs by more than 50% from that of the gridcell of interest within each window.

The preference for naive pooling does not hold in every context–shape-only pooling-like schemes have proven useful in other contexts[87,88]. Supplementary Fig. S3 shows that shape-only pooling does improve the fit for very high quantile levels, however, for our purposes the fit for moderate quantile levels is at least as important. Shape-only pooling reduces goodness of fit for the quantile level range in which most records lie, which is why is it unsuitable for our purposes. We are confident in the choice for naive spatial pooling since previous studies have also found naive spatial pooling to be a good method to improve GEV fits, despite its simplicity[39,89]. Fitting non-stationary GEV distributions to observations adds uncertainty relative to stationary ones. We nonetheless choose to do this, since climate change affects Rx1d in the historical period, leading to decreasing rareness of extremes. We verified that stationary GEVs overestimate record quantile levels for records that occurred recently, and underestimate corresponding record-breaking probabilities in the future. Differences between the two fits in earlier years are very small, indicating that the GEV fits do not become considerably different or worse due to the non-stationary fitting procedure. Lastly, Akaike and Bayesian information criteria are overall (marginally) lower for non-stationary fits for all three datasets, corroborating that non-stationary GEVs are preferred to stationary ones for the data at hand.

The accurate estimation of local (extreme precipitation) GEV distributions from observations is a very important pending issue that has been receiving attention recently, and still requires more. This paper's focus, however, is the assessment of regional disaster potential. We use the imperfect methods at hand, point out the limitations of our observed quantile level estimation, and leave the improvement of methods that can increase the accuracy of disaster potential quantifications for future research.

Although spatial pooling improves the observational GEV fit, considerable uncertainties still remain. Below, we quantify and compare different sources of uncertainty, and expand on the procedures used to produce the uncertainty ranges shown in Figs. 1 and 6.

**GEV distributions for simulation data.** Gridcell specific non-stationary GEV distributions are fitted to the Rx1d data of the eight CMIP6 models as well. For the 'true' full-ensemble GEV fit, all ensemble members (ranging from 5 to 50 for the different models, see Supplementary Table S1) and all years (1850–2100) are pooled per gridcell, utilising the maximum sample size to estimate the parameters. These GEV fits are not the true distribution of Rx1d in our real climate, but can, based on the large sample size, be assumed to be the true underlying distribution of the simulated Rx1d. For the model-based GEV fits, we use non-

stationary covariates comparable to the method used for observations, for tractability and robustness: both $g_\mu(t)$ and $g_\sigma(t)$ are the global ensemble mean surface temperature per model, smoothed with a loess filter.

The GEV distributions described above are used for the projection of future distributional changes. For validation and testing purposes, see Sect. 4.4, GEV distributions are fitted to single model members 'as observations' using the observational GEV method described above.

**Generalised extreme value distribution uncertainties.** The use of small samples in a GEV context requires thorough uncertainty quantification and communication to correctly interpret results and their confidence level. We do so by parametric bootstrapping, implying that we bootstrap the coefficients of the observational and reanalysis GEV fits. The bootstrapped GEV distributions are generated per gridcell by drawing $10^4$ random coefficient sets from a multivariate normal distribution, described by the five fitted GEV coefficients and their covariance matrix (output from the GEV fitting procedure). The covariance matrix of the fitted GEV coefficients reflects how well-constrained the maximum likelihood estimation used for fitting is, given the available data. The small observational sample size and variable Rx1d data are reflected in large covariances, which result in a wide range of bootstrapped GEV coefficients that span the uncertainty of the fit. Using the bootstrapped GEV coefficients sets, we determine $10^4$ quantile levels corresponding to the observed historical record value that describe the uncertainty range in observed quantile levels. We determine the lower and upper bound of the 95% confidence interval of these quantile level samples (2.5th and 97.5th percentile), and propagate these bounds through our analysis routines to represent the uncertainty range introduced by the observational GEV fits. The resulting uncertainty bounds in the analysis metrics of interest are shown in Figs. 1 and 6. The uncertainty bounds due to GEV uncertainty only in those figures (dark grey) correspond to the multi-model mean results for the 2.5th and 97.5th percentile record level inputs. The GEV uncertainty + model envelope (light grey) also include model differences, and show the model envelope spanning the lowest/highest single-model output for either of the record quantile level confidence bounds.

Figure 7d–f show the magnitude of the 95% confidence interval (CI) in the record quantile level, which can be compared to the actual record quantile level value in Fig. 7a–c. Clearly, spatial variability in the record quantile level is about as large as the uncertainty in individual gridcells, ranging up to 0.05. The uncertainty range is largest for HadEX3, as the resolution of HadEX3 is lower and spatial pooling windows are smaller. For all three datasets, the small sample size proves to be a major source of uncertainty, as also witnessed by the propagated uncertainties in our analysis metrics shown below and in Fig. 1 and 6. This epistemic uncertainty in observational GEV distributions is hard to overcome given that no additional observations exist. Efforts using weather model ensembles, such as UNSEEN[66,67], and potentially advancements in artificial intelligence for e.g., stochastic weather generators can be used to generate new 'observational' data, although validation remains difficult.

Assessing propagated uncertainties in CCP in 2050 and 2015 state likelihood shows that observational GEV fits are the largest source of uncertainty in the midlatitudes, while model uncertainties dominate in the tropics in the future, as shown in Fig. 8. The range spanned by multi-model mean CCP values corresponding to the 95% confidence interval bounds of observed record levels (a-c) is constant in space. HadEX3 has the largest uncertainty due to observational GEV uncertainties of the three datasets, corresponding to the larger uncertainty in record quantile levels. For CCP, GEV-induced uncertainty ranges are 0.4–0.5 for HadEX3 and around 0.3 for REGEN and ERA5. The multi-

model range (using the single best-estimate observed record quantile level) is around 0.1-0.2 in the midlatitudes for all datasets, but around 0.5 in the tropics (d-f). As 2050 CCP values are largely above 0.5 (see Fig. 3), this means reasonable signal-to-noise ratios are obtained in the midlatitudes, whereas in the tropics model uncertainty results in low confidence.

For state likelihood model differences are not shown: they are negligible since models have not diverged much by 2015. The state likelihood range due to GEV uncertainties (g-i) shows the same behaviour as CCP uncertainties, with slightly lower values as the uncertainties have not had as much time to grow.

The correlations between high (low) CCP in 2050 (state likelihood in 2015) and observational GEV-induced uncertainty ranges is positive, meaning that higher-risk regions are also subject to larger uncertainties resulting from the initial record quantile level uncertainty. Pearson correlations for all datasets are about 0.2 and 0.6 for CCP in 2050 and 2015 state likelihood, respectively. The correlation decreases with time, since more gridcells approach the probability asymptote of 1. This implies that uncertainties are largest for high near-term risk gridcells. This is in line with uncertainties being larger for lower record quantile levels.

Observational GEV uncertainties and model differences dominate the uncertainties; uncertainties in single climate model GEVs are consistently smaller since the models have multiple members.

## Record-breaking precipitation disaster potential indicators

The following section revisits and expands on the explanation of cumulative (conditional) record-breaking probabilities (CCP and MCP) and state likelihood in the main text. In the main text, we use two main indicators of record-breaking precipitation probability: the forward-looking cumulative record-breaking probability and the backward-looking state likelihood.

**Future cumulative record-breaking probability.** For future years, we define the future cumulative record-breaking probability. We separate the contributions of the two main drivers of record-breaking probability–natural variability and climate change–by determining the history-independent marginal cumulative record-breaking probability (MCP) and the history-aware conditional cumulative record-breaking probability (CCP), in counterfactual stationary and non-stationary climate conditions. Both MCP and CCP are functions of future year $N$. We often refer to its value in 2050, as a measure of the chances of breaking the precipitation record in the near term. We start the MCP and CCP calculations in the first unobserved year 2016, in order not to mix historical and future timeperiods. This means the record-setting year $T$ does not influence MCP and CCP, as $T$ is set to 2015 for all gridcells.

$$\begin{cases} \Pr_t(R) = \Pr(\mathrm{Rx1d}_t > \max(\mathrm{Rx1d}_1, \dots, \mathrm{Rx1d}_{t-1})) = \int_{-\infty}^{\infty} f_t(\mathrm{Rx1d}) \prod_{i=1}^{t-1} F_i(\mathrm{Rx1d}) \, d\mathrm{Rx1d} \\ \Pr_t(R|L) = \Pr(\mathrm{Rx1d}_t > L_m) = 1 - F_t(L_m) \end{cases}$$

(7)

$$\begin{cases} \Pr(Y_{\mathrm{NR}} = n | T = 2015) = \prod_{t=T+1}^{n-1} 1 - \Pr_t(R) \cdot (\Pr_n(R)) \\ \Pr(Y_{\mathrm{NR}} = n | L, T = 2015) = \prod_{t=T+1}^{n-1} 1 - \Pr_t(R|L) \cdot (\Pr_n(R|L)) \end{cases}$$

(8)

$$\begin{cases} \mathrm{MCP} = \Pr(Y_{\mathrm{NR}} \le N | T = 2015) = \sum_{n=T+1}^{N} \Pr(Y_{\mathrm{NR}} = n | T = 2015) \\ \mathrm{CCP} = \Pr(Y_{\mathrm{NR}} \le N | L, T = 2015) = \sum_{n=T+1}^{N} \Pr(Y_{\mathrm{NR}} = n | L, T = 2015) \end{cases}$$

(9)

Equations (7)–(9) show the intermediate steps for both the marginal and conditional probability metrics, where $\Pr(R)$ refers to the probability of breaking a record, and conditioning on the historical record $L$ is indicated. All $f(\mathrm{Rx1d})$ and $F(\mathrm{Rx1d})$ distributions

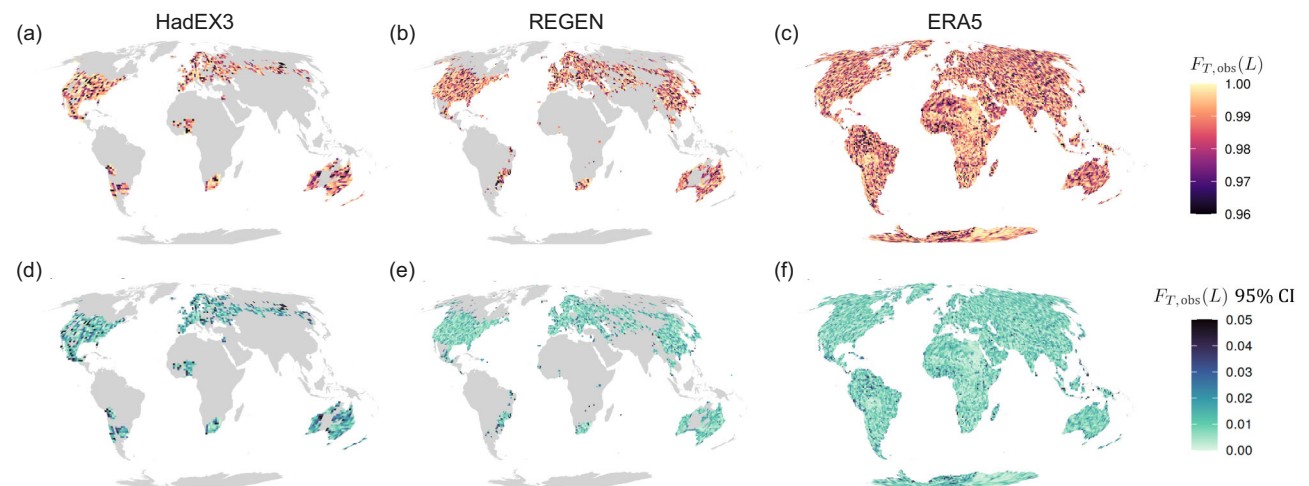

**Fig. 7 | Uncertainty in observed record quantile levels. a–c** Historical record quantile levels of record $L$ in year of occurrence $T$ according to fitted generalised extreme value distributions (GEV) (as in Fig. 4). **d–f** corresponding 95% confidence intervals of record quantile levels obtained through bootstrapping GEV coefficients $10^4$ times.

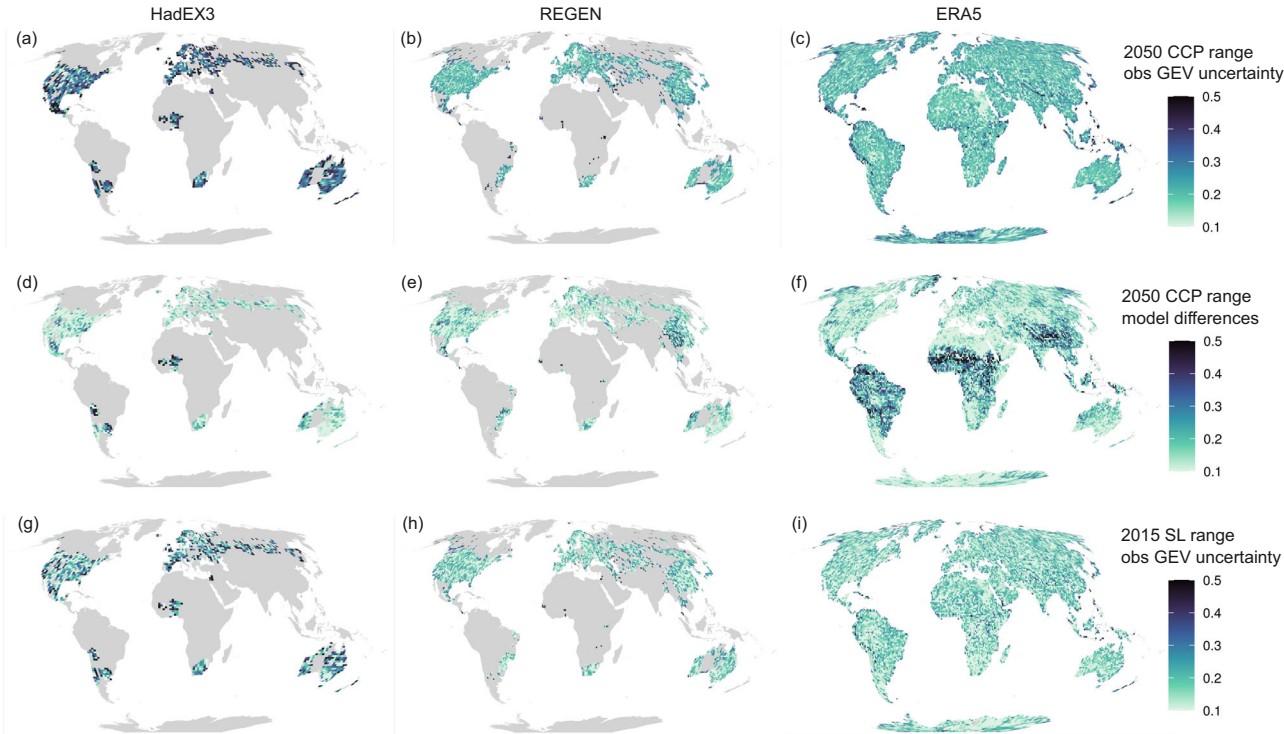

**Fig. 8 | Comparison of uncertainties introduced by observed precipitation distribution fits and intermodel differences. a–c** Observational generalised extreme value distributions (GEV) uncertainties in cumulative conditional record-breaking probabilities (CCP): Min-max range of multi-model mean CCP in 2050 obtained by bootstrapping GEV coefficients $10^4$ times and propagating the 95% confidence range of historical record quantile levels. **d–f** Model uncertainties: Min-max range of single-model mean CCP 2050 values resulting from across-model differences in climate change trends. **g–i** Observational GEV uncertainties in 2015 state likelihood (SL): Min-max range of multi-model mean 2015 SL values obtained by bootstrapping GEV coefficients $10^4$ times and propagating the 95% confidence range of historical record quantile levels. Uncertainty range due to model differences is comparatively negligible for 2015 state likelihood and not shown.

refer to the probability density function (PDF) and cumulative distribution function (CDF) of non-stationary GEV distributions fitted to climate model data (see previous section).

As evident from the equations, the MCP is the result one would obtain if one could average over infinitely many independent Rx1d timeseries for the same location and period. It is an aggregated metric over all possible local record quantile levels, and by definition independent of specific historical local record levels. As it is not

conditional on history, it is an unconditional probability and hence we refer to it as the marginal cumulative record-breaking probability (MCP).

The MCP is computed from the theoretical record-breaking probability as a function of time. In a timeseries where every value is drawn from of an i.i.d. (thus also stationary) sample, the marginal record-breaking probability as shown in equation (7) is independent of the underlying distribution and can be greatly simplified: the record-

breaking probability in the $j$th observation is is simply $\frac{1}{j}$[61]. In other words, for a large number of i.i.d. Rx1d timeseries, the fraction of timeseries with a record-breaking event at timestep $j$ will be $\frac{1}{j}$. In a timeseries where every value is drawn from of an independent but changing sample, the marginal record-breaking probability in year $t$ is given by equation (7)[60]. The integration over all possible values of Rx1d makes the resulting record-breaking probability independent of record quantile level.

In summary, the MCP is independent of the historical record level and defined as MCP($N$) := $P(Y_{NR} \leq N | T = 2015)$. We do not condition on the level $L$, thus, it is the aggregated probability over all possible Rx1d record levels $L$.

CCP is the cumulative record-breaking probability conditional on the record level $L$ to be exceeded. Since the record level is a result of natural variability, CCP includes the effect of natural variability on record-breaking. $F_t(L_m)$ in equation (7) refers to the time-varying CDF based on GEV distributions fitted to the CMIP6 models, evaluated at the quantile mapped record level $L_m$ (see below). The counterfactual stationary CCP (CCP in a stationary climate) is obtained by keeping the record quantile level in equation (7) constant in time, i.e., $F_t(L_m) = F_{T,obs}(L)$ for all $t$.

In summary, the CCP is dependent on the historical record level and defined as CCP($N$) := $\Pr(Y_{NR} \leq N | L, T = 2015)$.

Our computation of the conditional record-breaking probabilities (equations (8) and (9)) requires that the conditional record-breaking events in different years are independent of each other. This follows from independence of the year-to-year Rx1d values, or in other words, absence of autocorrelation in the Rx1d timeseries, shown in Supplementary Fig. S1. The computation of non-stationary MCP with equations (8) and (9) is approximate, since the marginal record-breaking probabilities in different years are no longer independent in case of a trend in the underlying distribution, even if the year-to-year Rx1d values satisfy independence. However, the differences between the approximated and the true probabilities would only be noticeable in case of a distribution shift of much larger magnitude than caused by plausible climate change. Thus, in all cases considered here, the above equations give a good approximation.

As mentioned, the stationary and non-stationary MCP and CCP enable us to separate natural variability and climate change effects on record-breaking probability. To give an overview, a summary of the metrics is given below, and Fig. 1 in the main text shows a visual example.

1. The MCP in a counterfactual stationary climate, with record-breaking probability at timestep $j = \frac{1}{j}$, i.e., the theoretical marginal cumulative record-breaking probability without climate change
2. The non-stationary MCP, with record-breaking probability at time $t$ given by equation (7), i.e., the theoretical marginal cumulative record-breaking probability with climate change
3. The CCP in a counterfactual stationary climate, with time-invariant record-breaking probability $F_{T,obs}(L)$, i.e., the cumulative record-breaking probability conditional on natural variability without climate change
4. The non-stationary CCP with time-varying $F_t(L)$ i.e., the cumulative record-breaking probability conditional on natural variability with climate change: the total CCP.

**State likelihood.** The backward-looking indicator 'state likelihood' is a combined measure of the current record quantile level and the time since this record was set – the record gap. The lower the current record quantile level and the longer the record gap, the lower the present state likelihood. This metric is a qualitative indication of local preparedness for unseen natural hazards and a quantitative indication of elevated present record-breaking probabilities, thus bearing relevance to the potential that extreme precipitation leads to a natural disaster. The state likelihood is the cumulative probability of not breaking the last observed record between its year of occurrence and the last observational timestep.

We use equations (7)–(9), but now $T$ is the last record-breaking year that set the current record and $N$ is 2015, the end of the observational record. The initial record quantile level is determined in year $T$ as $F_{T,obs}(L)$, where $F_{t,obs}(Rx1d)$ is the local CDF of Rx1d at time $t$, given by a non-stationary GEV distribution fitted to the observational/reanalysis Rx1d as described in the previous section. To determine the evolving record-breaking probabilities $F_t(L)$, we use the GEV distributions fitted to the multi-model ensemble data in the historical period. We map the observed record quantile $F_{T,obs}(L)$ onto the respective model distributions to obtain equivalent time-varying model record quantile levels $F_t(L_m)$, where $F_t(Rx1d)$ again refers to GEV distributions fitted to the climate model data. See the next paragraph for details on the mapping procedure. Even though we have observational GEV distributions for the historical period, we prefer using the model-based GEV distributions to inform $F_t(Rx1d)$ since these are more robust and more specific due to the larger sample sizes available for fitting, see Sect. 4.2. The difference in state likelihood values computed with model-based GEV distributions compared to observations-based GEV distributions is small.

Equation (9) shows the conditional cumulative probability of breaking the record $L$ before or in a given year $N$. State likelihood is defined as state likelihood (2015) := $1 - \Pr(T < Y_{NR} \leq 2015 | L)$.

**Quantile mapping.** The above explanation implies that translation of observed record levels to quantile levels in simulation-based GEV distributions is required for the determination of future cumulative probabilities and state likelihood. This is necessary since Rx1d distributions in physical units differ between observations and models, as well as across models. Translating Rx1d values to nondimensional quantile levels removes the influence of differences in absolute Rx1d values and enables the use of modelled distributions to project record-breaking of observed Rx1d levels. Importantly, this makes all metrics dependent on the modelled distributions, which may differ from observed/true distributions. The multimodel ensemble used helps address these uncertainties: differences in distribution shapes and climate sensitivity between models result in a range of future probability evolutions.

We translate the observed quantile levels as follows. We determine the quantile level $Q_{T,obs}$ of the record level $L$ at its time of occurrence $T$ from the observed CDF $F_{T,obs}(Rx1d)$. $Q_{T,obs}$ is used to determine the equivalent record level (in physical units) $L_m$ in models corresponding to this quantile level using the model-derived CDFs $F(Rx1d)$. This is shown in equation (10). In equations (8) and (9), $F_t(L_m)$ is used as the temporal evolution of conditional record-breaking probabilities.

$$Q_{T,obs} = F_{T,obs}(L)$$
$$L_m = F_T^{-1}(Q) \tag{10}$$

## Method validation
**Dataset agreement.** Given the reliance of our estimates on observations, we assess the degree to which the observed records and derived quantities are similar across the observational datasets and reanalysis data we use. HadEX3, REGEN and ERA5 do not show perfect agreement in their record evolution. Of the overlapping gridcells between HadEX3 and REGEN, about 30% exhibit the same last historical record-breaking year (after correction for different resolutions). For HadEX3 and ERA5 this is only 15%, and for REGEN and ERA5 about 25%. Across all three datasets, 8% of shared covered area exhibits the same last record year. Supplementary Fig. S2 shows the location of the gridcells with corresponding record-setting years $T$. The gridcells with corresponding $T$ do not follow any apparent pattern and are present in all covered regions.

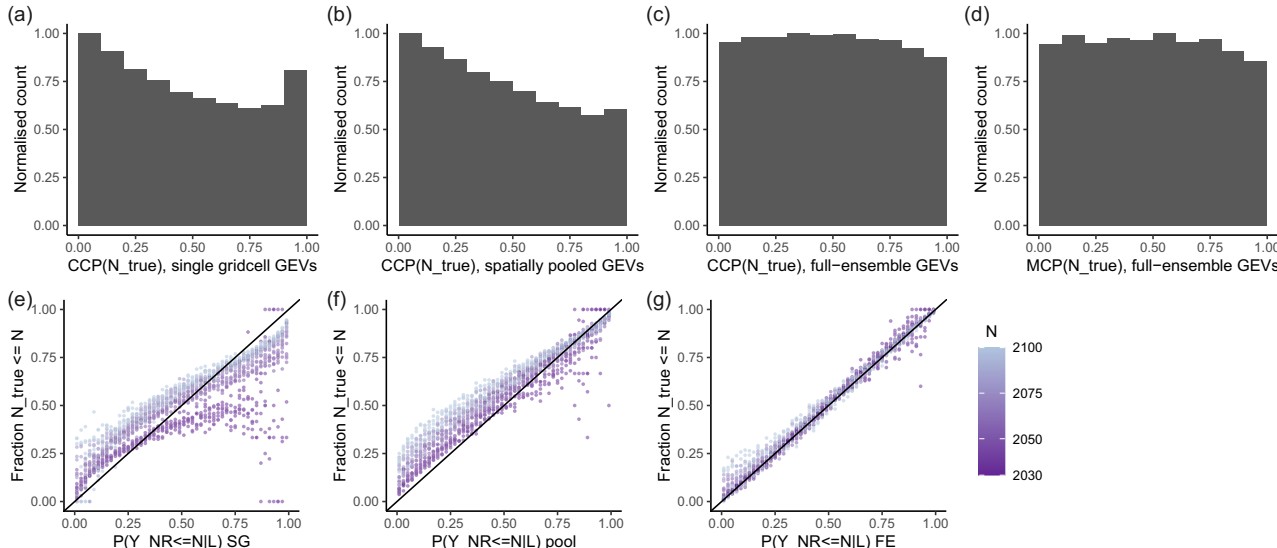

**Fig. 9 | Validation metrics for conditional cumulative record-breaking probability (CCP).** **a** Probability integral transform (PIT) histogram of the CCP if record quantile levels are determined relative to generalised extreme value distributions (GEV) fit to single gridcell `as observations' samples. **b** PIT histogram of the CCP if record quantile levels are determined relative to GEV distributions fit to spatially pooled `as observations' samples. **c** PIT histogram of the CCP if record quantile levels are determined relative to GEV distributions fit to the full model ensemble. (**d**) PIT histogram of the MCP. **e**–**h** Reliability of CCP (Pr($Y_{NR} \leq N | L$, $T = 2015$)) estimates evaluated in CMIP6 models on the HadEX3 grid.

**e** Historical record quantile levels used in equations (8) and (9) based on GEV distributions fitted `as observations' to single gridcell (SG) data. (f) Historical record quantile levels based on spatially pooled GEV distributions. **g** Historical record quantile levels based on the full-ensemble `true' GEV distributions (see 4.2 for details). The colour indicates for which year the reliability was evaluated, i.e., for 2050 we binned gridcells and members based on their Pr($Y_{NR} \leq 2050 | L$, $T = 2015$) CCP values and determined the fraction with $N_{true} \leq 2050$ for each CCP bin. y-values of 0 or 1 correspond to bins with only one or very few gridcells that all have the same record-breaking status.

For those gridcells with corresponding $T$ in pairs of datasets, we compute dataset correlations for CCP in 2050. The correlation of CCP in HadEX3 and REGEN is 0.27, for HadEX3 and ERA5 it is 0.20 and for REGEN and ERA5 0.23. The different nature of the observations and reanalysis is part of the explanation for the low agreement, as well as the different resolutions, which also strongly affect the GEV fit due to the pooling and thereby quantile levels and CCP. Differences between HadEX3 and REGEN are additionally influenced by the different gridding methods used[52,53].

**CCP and MCP evaluation.** The relevance of the state likelihood metric for disaster potential can be qualitatively validated with historical data, which we do in Sect. 2.4. MCP and CCP, defined in the future only, can be validated in models, which we do below. We treat climate model ensemble members as observations, and estimate member-specific MCP and CCP values for 2016–2100. Subsequently, we assess the agreement between the estimated probabilities and the actual future record-breaking behaviour in the model member.

This setup also allows us to assess the effect of the short observational samples. In the full-ensemble GEV setup, we determine the historical record quantile level relative to the full-ensemble GEV fit, which has higher accuracy due to higher data availability. We compare this to two, less accurate, 'as observations' setups. For the first one, the historical record quantile level is determined relative to a GEV distribution fitted to single gridcell data from single model members. For the second one, the historical record quantile level is determined relative to a GEV distribution fitted to spatially pooled data from single model members. The MCP and CCP values for the full-ensemble GEV setup are essentially the best we can do, and comparing the two 'as observations' MCP and CCP values allows us to assess the bias due to a small sample, and the improvement brought about by spatial pooling. Based on this comparison we chose to use spatial pooling for observations, see Sect. 4.2 and Supplementary Note S2. In the validation we focus on the model data regridded to the HadEX3 grid.

**Calibration.** We use probability integral transform (PIT) histograms (the continuous equivalent of rank histograms) to assess the calibration of CCP[90]. For each gridcell, each model member, and each GEV fit (single gridcell, spatially pooled, and full-ensemble) we computed a CCP value for each year $N$ between 2016 and 2100, which can be considered a probabilistic prediction, see main text and previous section. Each gridcell in each model member also has an actual next record year $Y_{NR} = N_{true}$, varying between 2016 and 2100, or unknown if the record is not broken by 2100. We call this the observation. PIT histograms assess the bias of predictions relative to the observations: in our case they are histograms of the CCP values (probability predictions) corresponding to the $N_{true}$ years (observations): CCP($N_{true}$).

In order to arrive at the PIT histogram, CCP($N_{true}$) is extracted for each gridcell in each model member. To prevent systematic overestimation of CCP($N_{true}$) related to the discrete cumulative probability distribution (one value per year) we select a value at a random location on the linear interpolation between CCP($N_{true} - 1$) and CCP($N_{true}$). We condition the PIT histograms on the true record being broken by 2100[91], since these are the only cases for which we can directly extract the corresponding CCP.

A prediction is considered calibrated if the PIT histogram is flat: this indicates that the CCP predictions follow the same distribution as the true next record year $N_{true}$, and do not systematically over- or underestimate $Y_{NR}$. Consequently, a PIT histogram that slants downward towards the right indicates that the observed values lie more to the left side of the prediction distribution, meaning that the prediction overestimates $Y_{NR}$ (predicts too late years). Conversely, if the PIT histogram slants upward towards the right, the observations lie primarily on the right side of the prediction distribution, meaning that the prediction underestimates $Y_{NR}$ (predicts too early years). If both these things occur on either side of the prediction range, a U-shape or inverted U-shape results. A U-shaped PIT histogram indicates too narrow a distribution of predictions relative to the truth, an inverted U-shape indicates too wide a prediction distribution[92].

**Table 1 | Continuous ranked probability scores (CRPS) and ranked probability skill scores (RPSS) for different conditional cumulative record-breaking probability (CCP) estimation methods**

|  | CRPS | RPSS |
| --- | --- | --- |
| CCP Single gridcell 'as obs' GEV | 0.19 | 0.05 |
| CCP Shape-only spatial pooling 'as obs' GEV, 3 × 3 window | 0.21 | −0.06 |
| CCP Naive spatial pooling 'as obs' GEV, 3 × 3 window | 0.17 | 0.13 |
| CCP Single gridcell full-ensemble GEV | 0.14 | 0.30 |
| MCP | 0.2 | 0 |

Both metrics are bounded between 0 and 1. A lower CRPS signifies a sharper, calibrated prediction, and the RPSS indicates the improvement in the prediction relative to a benchmark, the marginal cumulative record-breaking probability (MCP) in our case.

PIT histograms of the aggregation of all gridcells and model members from all models are shown in Fig. 9. We show the aggregated histograms as they look virtually identical when assessed on an individual model level. We show PIT histograms for non-stationary CCP values determined based on record quantile levels from single gridcell GEV, spatially pooled GEV, and full-ensemble GEV setups, and for the MCP, which is independent of record quantile level and hence also of GEV setup.

The PIT histograms for the single gridcell and spatially pooled 'as observations' CCP (Fig. 9a and b) are U-shaped, although primarily slanted downward towards the right. The predictions are thus too narrowly distributed: we primarily overestimate $Y_{NR}$ for lower CCP values (earlier years), and also underestimate the highest $Y_{NR}$ (high CCP values, rightmost bar). Spatial pooling solves most of the underestimation of late $Y_{NR}$, however, the overestimation of earlier years remains. This is caused by the too-narrow record quantile levels associated with the imperfect GEV fits due to small sample sizes, extensively discussed in Supplementary Note S2. Note that overestimation in our case means that we overestimate the year of next record occurrence (too late), and thus underestimate the cumulative record-breaking probabilities in any given year.

Figure 9 c is the main validation of our CCP estimation method, since this is the PIT histogram for predictions made using the full-ensemble GEV. We see a nearly flat histogram, indicating that our method is well-calibrated. However, some overestimation of the highest $Y_{NR}$ (underestimation of CCP values) manifests as the downward slanting towards the right. This bias primarily comes from members that break the record past 2050 (conditioning on 2050 produces an almost flat PIT histogram), and might therefore point to small errors that accumulate for later record-breaking years (which are associated with rarer events). Such small errors are likely due to biases in the full-ensemble GEV fit used to determine future record-breaking probability. It is a common issue that the true probability of very rare events is often underestimated by fitted GEV distributions[58]. Several factors could contribute to these errors. For example, 1-year block maxima of precipitation (Rx1d) are not truly max-stable, violating the assumptions for GEV fits; the maximum likelihood estimation used for GEV fitting introduces uncertainties[58,59]; and the non-stationary covariates which constrain the temporal evolution of the parameters might be biased due to smoothing or the small ensemble sizes of some models. We have looked into all these options and note that the results are indeed sensitive to such conditions and choices. Since there is no universal best practice regarding these questions, we do not aim to optimise the GEV fit for a perfect PIT histogram, but instead focus on the pragmatic solution where GEV fitting choices are intuitive and straightforward to implement.

Lastly, Fig. 9d shows the PIT histogram for the MCP, which is flat apart from a step change towards the right, most likely due to biases in the full-ensemble GEV fit, mentioned above. As the MCP is the marginal cumulative probability and is not sensitive to quantile level estimation errors, it is not surprising that this prediction is calibrated, yet an important validation.

To gain more insight into the reason for the PIT histograms' shapes, we assess reliability plots showing the fraction of observed model members and gridcells that has broken their record by a given year, binned per CCP-value. In case of a perfect prediction, the CCP value for a given year (probability prediction) would be equal to the fraction of members that breaks their record by the respective year (true probability). For example, CCP values of 0.4 by the year 2030 are estimated for an array of members and gridcells. To assess the reliability of the CCP estimates, we quantify which fraction of ensemble members × gridcells with a CCP of 0.4 by 2030 in fact breaks the record by 2030. If the CCP prediction were perfect, this fraction would equal 0.4. In Fig. 9e–c we show the results for the CCP estimates based on single gridcell, spatially pooled, and full-ensemble GEV fits, respectively. We show the results for four different years N, namely 2030, 2050, 2075 and 2100 (colour of the markers), to evaluate dependence of CCP reliability on $Y_{NR}$. The fractions are computed for each model separately, in the figure all model results are plotted.

Figure 9 e shows clearly how the CCP based on single gridcell ('SG') GEV fits underestimates the true record-breaking probabilities in the left half of the plot, where record-breaking fractions exceed the CCP values, and overestimates them in the right half, where fractions lie below the CCP values. This corresponds to the U-shaped PIT histogram: regions with high record-breaking probability are assigned a too low CCP, and those with lower record-breaking probability are assigned a too high CCP. This behaviour is seen for all years N, but as we move to later years, more and more gridcells have broken their record and we see that the underestimation increases (left half). Note that some of the spread in the right half of Fig. 9e is due to the low number of gridcells featuring high CCP values in 2030, exacerbated by the too narrow quantile level distributions in the SG-case. The true record-breaking probability follows the CCP estimates resulting from historical record quantile levels based on spatially pooled ('pool') GEV fits more closely (Fig. 9f), and most of the overestimation in the right half of the plot has been resolved. An underestimation of the true record-breaking probability at lower CCP values is still visible though, again increasing as N approaches 2100.

Figure 9 g shows how well aligned the CCP estimates and the true record-breaking probabilities are if record quantile levels can be determined accurately using the full-ensemble ('FE') GEV fits. Nonetheless, a slight underestimation in this prediction is visible in the points lying systematically slightly above of the 1:1 line, and again, the underestimation is highest for low CCP values and late years N. These plots are in agreement with the PIT histograms and show that the underestimation of true record-breaking probability is largest for lower CCP values (below 0.5) and late years N.

**Ranked probability skill scores.** The continuous ranked probability score (CRPS) is a combined measure of the sharpness, or specificity, and the calibration of a prediction. It simply consists of the summed squared error between the prediction and the truth: the smaller this number, the closer to and the more narrowly distributed around the truth the prediction is, and thus the sharper the prediction. Note, if the prediction is not calibrated, its CRPS will be high (bad) regardless of sharpness, since the offset with respect to the truth will be large. Again, the prediction is the CCP or MCP assigned to a year N, and the truth is the actual next record-breaking year $N_{true}$.

To compute the CRPS, $N_{\text{true}}$ is translated into a binary variable with a value for each year $N$: 0 for all years before $N_{\text{true}}$ and 1 for $N_{\text{true}}$ and all years after. The squared difference between the CCP or MCP and this binary variable summed over all years $N$ constitutes the CRPS[93]. Table 1 shows the CRPS values for the CCP/MCP estimation methods using different GEV fits for the historical record quantile, as above. The values lie closely together, but given that the prediction lies between 0 and 1, a difference of 0.02 in the CRPS does indicate that the uncertainty range of the prediction gets considerably wider.

The CRPS allows us to quantify the skill gained by different predictions relative to a benchmark. We consider the MCP our benchmark: this prediction does not take into account any gridcell specific information regarding the exceedance threshold for a record and can thus be considered the climatological value. It is centred around the correct value (calibrated, as we saw above), but not sharp at all (wide distribution). We compute ranked probability skill scores (RPSS) as $1 - \frac{\text{CRPS}_i}{\text{CRPS}_{\text{MCP}}}$, where $\text{CRPS}_i$ refers to the CRPS of the different CCP methods and $\text{CRPS}_{\text{MCP}}$ is the benchmark CRPS. The RPSS ranges from $-\infty$ to 1, where any value $< 0$ indicates the prediction is worse than the benchmark, 0 indicates the prediction adds no improvement relative to the benchmark, and any value between 0 and 1 indicates improvement, with 1 being a perfect prediction that is always correct.

Table 1 shows RPSS values for the CCP estimates resulting from the indicated GEV fitting methods. CCP values based on full-ensemble GEV fits, where all model information is used to fit GEV distributions, yield a 30% smaller CRPS relative to the benchmark MCP, indicating a significant improvement in sharpness of the estimates. From the 'as observations' ('as obs') options, naive spatial pooling results in the largest improvement relative to the benchmark. Supplementary Note S2 on spatial pooling discusses the clear superiority of $3 \times 3$ spatial pooling (at HadEX3 resolution) over single gridcell or shape-only pooling GEV fits for the observational CCP estimates. These results, in combination with all validation results shown above, confirm the validity of our method to estimate future cumulative record-breaking probabilities, conditional on historical observations. They also show that the short observational timeseries have a major negative influence on accuracy of real-world CCP estimates. Our estimates are too narrow, implying both under- and over-estimation of true probabilities at individual locations. Aggregated over the globe, our estimates underestimate the true probability of record-breaking, which necessitates caution in interpretation, even more so when using such estimates in downstream risk assessments.

## Data availability

All original CMIP6 data used in this study are publicly available at (https://esgf-node.llnl.gov/projects/cmip6/). HadEX3 and REGEN data are publicly available on (https://www.climdex.org/access/) ERA5 data are publicly available from the Copernicus Climate Data Store at (https://cds.climate.copernicus.eu/#!/home). Processed data as shown in the figures in this paper are available at (http://hdl.handle.net/20.500.11850/785553). Additional data is available upon request.

## Code availability

The code to perform the analysis and produce the figures in this paper are available at (http://hdl.handle.net/20.500.11850/785553) additional code is available upon request.

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

## Acknowledgements

We would like to thank Sam Allen, Lukas Graz, Jonas Peters, Daniel Raimundo, Mark Risser, Joel Zeder, and Wim Thiery for helpful input. We thank Urs Beyerle, Lukas Brunner and Ruth Lorenz for the preparation and maintenance of CMIP6 data, Martin Hirschi for managing the ERA5 data used in this study, and Mathias Hauser for assisting with data and programming issues. We acknowledge the World Climate Research Programme's Working Group on Coupled Modelling, which is responsible for CMIP, and we thank the climate modelling groups for producing and making available the model output. For CMIP, the U.S. Department of Energy's Programme for Climate Model Diagnosis and Intercomparison provides coordinating support and led development of software infrastructure in partnership with the Global Organisation for Earth System Science Portals. We acknowledge CLIMDEX (www.climdex.org) for making available the observational datasets and extreme indices used in this study, and the European Centre for Medium-Range Weather Forecasts (ECMWF) for providing ERA5 reanalysis data. The analysis was carried out in R (R Core Team, 2022), we thank all contributors for the numerous R-packages crucial for this work, in particular the extRemes package[82] and the maps package[94]. EF was supported by funding from the EU Horizon 2020 Project XAIDA (grant no. 101003469).

## Author contributions

I.V. contributed to conceptualisation, method development, method implementation and data analysis, and writing and visualisation. M.S. contributed to conceptualisation, method development and text improvements. E.F. and Se.S. contributed to conceptualisation and text improvements, and RK contributed to conceptualisation, text improvements and funding acquisition.

## Funding

## Competing interests

The authors declare no competing interests.
