## [Transparent Peer Review file · Nature Communications]

Precipitation disaster hotspots depend on historical climate variability

Corresponding Author: Dr Iris de Vries

Version 0:

Reviewer comments:

Reviewer #1

(Remarks to the Author)

- What are the noteworthy results? Perhaps
- Will the work be of significance to the field and related fields? How does it compare to the established literature? If the work is not original, please provide relevant references. It does not account for established literature and is not state-of-the-art.
- Does the work support the conclusions and claims, or is additional evidence needed? Not convinced.
- Are there any flaws in the data analysis, interpretation and conclusions? Do these prohibit publication or require revision? Difficult to say because it was a hard read and made the analysis much more complicated than necessary
- Is the methodology sound? Does the work meet the expected standards in your field? I doubt it is sound. No, it does not meet the expected standards.
- Is there enough detail provided in the methods for the work to be reproduced? It would be a tough job to reproduce these results.

See attached PDF-file with more detailed review

[Editorial Note: See end of file]

(Remarks on code availability)

NA. The manuscript was so poor that I didn't see the point of reviewing the code.

Reviewer #2

(Remarks to the Author)

- The paper presents the analysis of record-breaking precipitation probabilities and their potential of high-impact disaster in future climates, at global scale. The analysis is developed by conditioning the record-breaking probabilities on historical observations, and considering climate change according to a large ensemble of global climate models. The disaster potential is then analyzed based on those probabilities, assuming that in case of low records and/or no recent records the preparedness for disaster risk could be lower, thus the disaster potential higher.
- The topic is relevant, with novel aspects with respect to more usual analysis of changes in return levels. The results are interesting and relevant, and conclusions are supported by the analysis. I think the paper meets the scope of the journal. Anyway, I found not easy the reading, and to understand the meaning and implication of the several probabilities presented. I suggest minor revisions (listed below), mostly with the aim to enhance clarity and readability of the paper.
- Section 2.1. I suggest to help the reader by linking better the textual explanations with lines in figure 1b,c, because those probabilities are better clarified and could be better understood just later in the methodology section. For example, 1) add "non stationary conditional" at line 155 after Fig1b. 2) write "conditional" cumulative record breaking probability at line 162. Also, maybe add a short explanation of what the 4 lines shows (natural variability, climate change, both, nothing). This is better explained at section 2.3, but it is not easy to understand this in section 2.1
 - Figure 1b,c: it is difficult to distinguish the 4 lines in the light yellow color. Maybe a darker one?
 - Line 158-159. From where do you see this? By comparing $P(Y_{nr=n|L})$ with stationary $P_t(R|L)$?
 - Line 192. Why is MCP "moderately" sensitive to climate change, considering that it is (line 187) "only sensitive to climate

change”?

- Section 2.2. at the beginning, I suggest to mention the different spatial coverage of the 3 observational datasets, to help the reader in interpreting fig2 (mostly grey for HadEX3 and REGEN).
- Lines 262-265. How do we see this from figure 3? How to look at natural variability vs climate change is just introduced later at lines 279-285...
- Lines 279-285. I suggest to present the ratios with the same order they appear in figure3 (figure 3b first, then 3c). This could help the reader interpreting text together with figures.
- Line 283. Stationary MCP?
- Line 303. Their stationary counterpart ...is the dotted line in fig1b? add in parenthesis “(dotted)”.
- Figure 3, caption for d) and e). Clearly indicate which lines show each effect (climate change, natural variability, their combination).
- Lines 461-462. What about doing for high state likelihood cells an analysis similar to that carried in this section for the low state likelihood cells?
- Line 621. 251 years ... which is the period for the models?
- Line 950 and 952 are a repetition of the same concept.

(Remarks on code availability)

Reviewer #3

(Remarks to the Author)

Review of “Where and when will the next record-breaking precipitation disaster occur?”

This manuscript proposes a novel framework to assess the probability of future precipitation records being broken, based on the current record value and its position within a fitted climate distribution. The central idea is that regions with seemingly low historical precipitation records may, under climate change, be at disproportionately high risk of record-breaking events — a concept tied to what the authors define as “state likelihood.” The methodology is validated in model space using CMIP6 large ensembles, and then applied to real-world observational datasets using spatially pooled GEV fits.

The paper is technically ambitious, innovative, and addresses a societally relevant question. However, I have concerns about the robustness of its conclusions — especially when applied to sparse observational datasets — and about the reliance on GEV fits as a tool to infer probabilistic extremeness. These concerns are outlined below.

Major Comments

1. Scope and fit

The manuscript introduces a promising framework and addresses a societally relevant question with clear implications for climate risk assessment and preparedness. While the broader significance is articulated, it is embedded within a presentation that leans heavily on technical formulation and statistical diagnostics. This places the manuscript toward the more methodological end of the spectrum typically seen in Nature Communications, which prioritizes conceptual advances of broad interdisciplinary appeal. The paper could be a strong fit for the journal if some of the technical exposition were streamlined to enhance accessibility to a broader readership.

2. Overreliance on GEV fits with limited data

While I appreciate that the manuscript acknowledges many of the challenges involved in fitting GEV distributions to short observational records and makes a reasonable case for using spatial pooling to stabilize fits, I remain concerned about the degree of reliance placed on GEV-derived quantile estimates for drawing key conclusions.

A central premise of the manuscript is the use of GEV-fitted distributions to assess the quantile level of current records and estimate future record-breaking probabilities. While I appreciate the careful validation in model space and the use of spatial pooling to improve stability, I remain concerned about the degree of trust placed on GEV fits, especially when applied to relatively short observational records or pooled data. As is well known, the estimation of the shape parameter ξ and the resulting quantile behavior are highly uncertain under these conditions.

This is particularly relevant when the conclusions hinge on determining that an observed record (e.g., in 2015) is “low” — meaning that it lies at a modest quantile (e.g., 90th percentile) of the fitted distribution, and is therefore likely to be broken. I find it problematic to fit a distribution to a small sample and then use that same distribution to assess how extreme the largest value in the sample is. This risks circular reasoning, and the inferred quantile level of the record can be highly sensitive to assumptions about the tail behavior (especially when $\xi > 0$).

I encourage the authors to either:

- Quantify the uncertainty of these GEV fits explicitly (e.g., using bootstrapping or ensemble spread),
- Or temper the strength of their conclusions when applied to real-world data.

A non-parametric benchmark — such as comparing against empirical quantiles or resampling-based assessments — would go a long way in grounding the results and offering a sense of robustness.

3. Inference that current records are “low”

Following on the point above, the paper’s logic assumes that one can reliably assess whether a current record is “low” (i.e., not unusually high relative to the underlying or shifting distribution). But how can one assert that a record is low using the same data that generated it?

The fitted GEV distribution — often based on only ~66 annual values per gridcell and spatial pooling — is then used to determine that this record lies at, say, the 90th percentile. But this percentile is itself derived from the same sparse dataset, via a model with strong assumptions. It is not surprising that such fits would result in record values being labeled “moderate,” particularly when the shape parameter allows for heavy tails. In many situations, I would argue the empirical distribution is more trustworthy than a fitted model with weak constraints.

In this light, the declaration that a record is “low” and that disaster potential is high seems overly confident. I strongly encourage the authors to discuss the possibility that GEV fits may overestimate the tail behavior, and that conclusions derived from these quantile assessments should be treated with appropriate caution.

4. GEV fitting under non-stationarity and compounding uncertainties

The idea of fitting time-varying GEVs to account for trends in extremes is, in principle, well motivated and aligns with current best practices in the field. However, given the limited length of observational records (~66 years), I believe even stationary GEV fits are already subject to substantial uncertainty. Introducing time dependence in μ and σ increases model flexibility but also compounds that uncertainty — especially when applied locally. While spatial pooling helps with stability, it also imposes smoothness assumptions that may not hold in topographically or climatologically complex regions. Overall, I find the approach conceptually strong, but optimistic in its application given the data constraints. I encourage the authors to clarify how these layered uncertainties propagate into the final probability metrics (e.g., CCP, MCP, and state likelihood), particularly in the observational analysis.

Minor Comments

L24–26: If I’m understanding correctly, this sentence may conflate current absolute risk with the rate of increase in risk. I recommend clarifying that regions with low historical records face high absolute record-breaking probabilities, whereas regions with high records may see the sharpest relative increases in risk due to climate change.

L47–49: The statement “extreme precipitation increases approximately exponentially with global warming” could be made clearer. Which aspect of precipitation extremes increase exponentially with global warming? Is it intensity (weakly nonlinear) or frequency (highly nonlinear and depending exponentially on the local Clausius-Clapeyron relationship; e.g., see Martinez-Villalobos & Neelin 2023, equation 3 or 15, <https://www.nature.com/articles/s41598-023-32372-3>). Please clarify.

L128: The GEV framework includes several layers of uncertainty, especially when fitting time-dependent parameters. How are these propagated into the final probability estimates? (see my major comments)

L137–138: The illustrative regions shown in figures are interesting, but how geographically representative are they? Were these selected a priori, or because they yielded good signal?

L217–220: Attribution to localized convective systems may be an oversimplification — such storms are often embedded in broader synoptic patterns. Please consider revising.

L262–268: How does internal variability influence the results? For instance, are different ensemble members consistent in identifying “low-record” locations?

L351–352: “The sections above showed...” seems too strong. Much of the analysis supports the idea that conditioning on local historical records may be important, but this is more a hypothesis than a conclusion fully demonstrated.

L364–366: The sentence implies a functional relationship between state likelihood and the discrepancy between actual risk and preparedness. It appears that state likelihood is being used as a statistical proxy for societal awareness or preparedness — a creative and potentially useful framing. However, while the narrative about awareness biases and lack of experience is plausible and well discussed, this connection remains conceptual and is not quantitatively evaluated in the manuscript. Is a linear or threshold-based relationship assumed? If not, I suggest clarifying that this is an interpretive use of the metric, not a demonstrated empirical link.

This is an interesting and timely paper that introduces a creative approach to diagnosing regional vulnerability to record-breaking precipitation extremes. The use of model ensembles for method validation is well executed, and the concept of state likelihood is clearly defined and tested. However, the application to observational data, and the inference that current records are “low” based on fitted GEV distributions, raise statistical concerns that warrant further attention. The conclusions, while potentially impactful, should be more cautiously framed, and uncertainties more clearly communicated. With these revisions, the manuscript could make a valuable contribution to the climate extremes literature.

(Remarks on code availability)

Version 1:

Reviewer comments:

Reviewer #2

(Remarks to the Author)

My previous comments were well received, and in my opinion, the other reviewers' comments were also quite satisfactorily answered. The topic is relevant, but the article is not easy to read, because of the complex methodology and complexity of results. My suggestion is for acceptance, but having in mind that is probably a paper at the limit of the "broad audience target" of the journal.

(Remarks on code availability)

not clear to me how to reach the "ETHZ repository – link TBA" indicated in the Code and data availability section

Reviewer #3

(Remarks to the Author)

I thank the authors for the substantial effort invested in this revision. The manuscript has improved considerably after this round of changes. While I share, to some extent, Reviewer's 1 concern that the literature review feels somewhat selective and omits several important references, I do not consider to severely undermine the quality of the work. In my view, the paper is strong, and I'm pleased to recommend it for acceptance.

(Remarks on code availability)

Reviewer #1 (Remarks to the Author):

- What are the noteworthy results? Perhaps
- Will the work be of significance to the field and related fields? How does it compare to the established literature? If the work is not original, please provide relevant references. It does not account for established literature and is not state-of-the-art.
- Does the work support the conclusions and claims, or is additional evidence needed? Not convinced.
- Are there any flaws in the data analysis, interpretation and conclusions? Do these prohibit publication or require revision? Difficult to say because it was a hard read and made the analysis much more complicated than necessary
- Is the methodology sound? Does the work meet the expected standards in your field? I doubt it is sound. No, it does not meet the expected standards.
- Is there enough detail provided in the methods for the work to be reproduced? It would be a tough job to reproduce these results.

See attached PDF-file with more detailed review

Reviewer #1 (Remarks on code availability):

NA. The manuscript was so poor that I didn't see the point of reviewing the code.

Review of 'Where and when will the next record-breaking precipitation disaster occur?' by de Vries et al.

In general, I find this manuscript a bit disappointing and suggest rejection. Below are some comments on which my evaluation is based.

We are sorry to hear that the reviewer was disappointed. Below we have tried to reply to the comments as well as possible, and we have made changes to the manuscript that hopefully address some of the issues and concerns the reviewer raised. In our responses below, all line numbers refer to the latexdiff file, since that makes it easier to see how the changes fit in the context.

I found the manuscript a bit disappointing in terms of acknowledging past research on record-breaking events, and it's important to account for relevant work and publications to give the reader some assurance that the authors are so well informed of the topic that it's worth spending the effort to read the paper. Furthermore, in science it's expected that colleagues are recognised for their work, but a lack of due literary research may also potentially indicate a biased presentation. I must admit that I sometimes wonder if my peers actually read the literature, as it's quite typical that relevant papers are not cited but a few are cited multiple times from within a smaller part of the community. At least, it's not a bad idea to do a thorough literary research and provide a comprehensive account on the topic of record-breaking statistics within climate research to avoid

glaringly different stories on record-breaking precipitation (... and if they do differ, then the differences should be explained). Also, some journals limit the number of self-citations to a small number of studies.

We fully agree that thorough literature research is essential, as well as citing and acknowledging previous work that has made the present research possible. We most certainly try to keep up with the literature on precipitation as well as extreme statistics, but are fully aware that we might have missed some things. We do not find that our results disagree in any way with previous research or theories on the matter, and we also do not aim to disprove any work that has been done on the topic. Regarding self-citations, thanks for making us aware of the fact that some coauthors do show up in the reference list rather often, however, relative to the total number of references it seems acceptable still. It happens to be so that the topic of this work is the expertise of at least one of our coauthors, and their involvement in the manuscript naturally impacts the line of thinking to be an extension of work they have already done. We choose not to remove any of these citations, since they are not performative, but we acknowledge that there are more authorities in the field, and we have read the papers the reviewer pointed us to, and added citations and references to this work where relevant.

The statement "*The high spatiotemporal variability of precipitation implies that events that are much more extreme than any previously observed event can occur, catching society by surprise*" may not be correct, as explained in DOI:10.1126/sciadv.ado3712, which showed that there have been more record-breaking temperatures (i.e. more extremes) in the tropics with smoother variations than in the high-latitude with more abrupt and pronounced variations. Also, this statement neglects the fact that the duration of the streaks with near-record-breaking weather may be just as damaging as the magnitude of more ephemeral events.

We appreciate this critical evaluation of the statement we made. Indeed, there is a lot more complexity to record-breaking than just "high variability means high record-breaking". In this sentence, we rather aim to describe a general experience that appeals to a broad audience, and do not wish to go into the weeds of the statistics yet. The rest of the paper features a thorough statistical analysis that takes into account the full complexity of the distribution, including the complex effects of variability on record-breaking over time.

We'd like to go into this a bit more. Record breaking in year Y requires two things: 1. A high value in year Y and 2. NO high values previously. This is where variability matters: high variability makes 1 likely, but makes 2 less and less likely as more time has passed before year Y. In addition, low variability makes mean trends more potent, as their signal to noise ratio is higher. Warming in the tropics, where temperature variability is low, is thus likely to emerge rather quickly in the form of systematically exceeding previous temperatures. Mean precipitation trends, on the other hand, will not emerge from the natural variability in precipitation easily. High precipitation variability, however, does still stand a chance of causing unprecedented events, given our short observational record. This is all to say: yes, it is complicated and variability is not the only thing that matters.

Yet, at this point in the manuscript, we think the experience-centric view is helpful, and we leave the statistics for later. We also refer to Fischer et al. (2021) and De Vries et al. (2024), where this mean vs. variability effect, and how temperature and precipitation differ, is discussed in more detail.

Lastly, we fully agree with the reviewer that record-breaking is not synonymous with impacts, and that non-record breaking events can also have high impact. We have added:

“We note that record-breaking events do not always have high impacts, and high impacts can also result from non-record-breaking events. While other definitions of extremes and high-impact events deserve equal attention, we focus here on record-breaking events, given their clear statistical and conceptual definition and their intuitive connection to unpredictable impacts.” (L79-82)

to the intro, to ensure this is communicated clearly.

The sentence *“The high spatiotemporal variability of precipitation implies that events that are much more extreme than any previously observed event can occur, catching society by surprise”* can also be a bit misleading because spatial/geographical variations of weather statistics are often well-known and highly predictable. It is mostly the temporal variability that provides surprises.

We see the point of the reviewer. We refer to both the spatial and the temporal variability here with the rest of the paper in mind: in regions with a certain climatology (that is well-known and predictable), some locations might still be subject to decades of low precipitation, while neighbouring locations might experience 3 deluges in the same period. Simply by chance: both regions are interchangeable in terms of true climatological distribution, but rain showers can be very local because of the large spatial gradients associated with precipitation events. This is indeed variability in time (if we wait long enough, both locations will experience the same amount of deluges) but in short time windows this manifests as different time-evolutions of extremes per location. Anyway, to avoid confusion in either direction we have removed the word “spatiotemporal” because it is not supposed to be the focus point of the sentence and does not add any essential information.

The sentence *“Extreme precipitation increases approximately exponentially with global warming, with a rate of 7% per degree Celsius of global warming [3, 13, 18, 24], but more extreme events see relatively stronger increases [12, 14, 25, 26]”* and the whole paragraph ignores studies which indicate that also the fraction of Earth’s surface area receiving daily rainfall is a factor determining the extremity of precipitation (see e.g. DOI:10.1038/s41612-024-00794-z and references therein). I think it’s important to be

up-to-date regarding the science in order to convince the readers, and hence, it's important to acknowledge also recent progress, even if it's an effort to keep abreast.

We respectfully disagree with the suggestion to include information on the fraction of the Earth's area receiving daily rainfall. Albeit we think this is an important and interesting finding in its own right, we do not think it is pertinent in this paragraph. The scope of the paragraph is intentionally kept narrow, to set a simple expectation baseline for extreme precipitation changes without elaborating on physical processes and redistribution of rainfall in space and/or time. The fact that we cannot and do not cite all the papers that looked into some aspect of extreme precipitation, does not mean that we are not up-to-date or deliberately not acknowledging certain studies. That said, we are very grateful for the reviewer to point us to related studies we had not yet included at the time of submission, but have now.

The passage "*These so-called record-shattering events [22, 23] – large jumps in record values – are rare, but can occur in the absence of strong climate change, due to natural variability alone*" this is only true for long time series, but not so for short series when it's more typical with large jumps in record-breaking levels. Also, it really depends on the underlying type of distribution of the variable, and we expect different step changes if it were normally distributed or followed some gamma distribution. Rather than citing their own work in this paragraph, they could provide a broader account of record-statistics such as the one in DOI:10.1126/sciadv.ado3712.

We see how this sentence might have skipped a few essential steps. We have changed it to:

"Record-breaking and so-called record-shattering events -- large jumps in record values -- become less likely as observational timeseries length increases, but remain possible due to natural variability even in a stationary climate." (L47-51)

This is still not a full description of the statistical intricacies of record-shattering in a non-stationary context, which is why we refer to Fischer et al. (2021) and De Vries et al. (2024), since these papers do provide a quite comprehensive statistical deep-dive into record-shattering.

Regarding the impact of distributions: we are always talking about annual records, meaning we are in the block-maxima space which can be considered to fall under extreme value theory and follow generalised extreme value distributions. Record breaking is independent of the underlying distribution (in a stationary context). Record shattering is more sensitive to the distribution given that some record-margin has to be exceeded for an event to be record-shattering. Explaining all this at this point in the introduction, whereas the focus of the paper is not on the statistics of record-shattering, was not deemed helpful. We now also cite the paper the reviewer suggested, and continue to cite Fischer and De Vries in this paragraph too, exactly because we do not explain every detail, but instead refer to places where more details can be found for interested readers.

I don't understand the point made by this sentence: "*Marginal record-breaking probabilities can be assessed by averaging many climate realisations [22, 23, 27], but local, real-world record breaking probabilities are dependent on the locally observed current record*" and I'm not sure that the statement is correct. Record-breaking probabilities usually are not estimated through taking averages and they vary from place to place and with the number of measurements taken. This is explained in literature not cited by de Vries *et al.* Also, future record breaking probabilities have been addressed before and there are published papers on the same topic as presented in this manuscript.

This sentence introduces the fundamental point of the paper. Marginal record breaking probabilities differ greatly from conditional record-breaking probabilities that take into account the actual standing record. The paper expands on this sentence and shows it is correct.

We have expanded this statement now to:

"Record-breaking probabilities are commonly assessed in a marginal, history-unaware manner by averaging many climate realisations, aggregating in space, or determining marginal probability distributions. In our more experience-centric approach to record-breaking events, however, we focus on the path-dependent probability of breaking the local observed standing record by conditioning on historical observations." (L71-76)

For additional context, we cite a few more papers that assessed marginal probabilities, including Benestad (2004) and Benestad *et al.* (2024). We also cite a few additional, very recent papers that came out after our initial submission, that took an approach more similar to ours. The review by Fischer *et al.* (2025) that also came out after initial submission, recaps how most literature focuses on marginal probabilities (apologies, that's another citation to a coauthor's paper, which cannot really be prevented given that this coauthor is an expert in the field).

What's the justification for the following statement: "*The observed local historical Rx1d record is associated with a certain quantile (percentile) level in the local Rx1d distribution, which defines the instantaneous record breaking probability*" and how do certain percentiles define record-breaking probability? And why call it "instantaneous"? Record-breaking probability is always defined by a collection of data and changes over time in a chronological fashion, as discussed previously in the literature.

There is no contradiction between our statement and the reviewer's view; the difference is that we refer to conditional probabilities: breaking the actual record, whereas the reviewer refers to marginal probabilities. The entire paper is focused on the concept of conditional probabilities versus marginal probabilities, we think that section 2.1 makes the concept very clear.

Just for additional explanation, a concrete example here, in a stationary context for simplicity. At one specific location (station, or gridcell) at location i , the probability of a record breaking Rx1d value is $1/n$ with n being the n th observed year. This changes over

time because n increases. This **marginal** probability calculation assumes that we **do not know** what happened in all the years n . If, however, we look at all $Rx1d$ values recorded at location i , and we see the record is 100 mm, then the **conditional** record breaking probability (conditional on the observed record), equals the probability of getting >100 mm rain. The probability of getting >100 mm rain, is $1-F(100)$ with F being the CDF of $Rx1d$ at location i . This value can be very different from $1/n$, and varies with F in a non-stationary context. This is all explained in section 2.1.

The word instantaneous has fallen out of the sentence in the revised version, but was referring to a non-stationary context; as F shifts with climate change, the probability of breaking the record increases even though the record to be broken (100mm) remains constant (until it is broken).

The sentence “For four single gridcells with all possible combinations of low/high record quantile level and weak/strong climate change, we show an example of the analysis metrics in Fig. 1.” is confusing. What is meant by “low/high record quantile level”. It seems like records (measurements) are mixed up with records (highest or lowest recorded value). Isn't a record the same as q_1 (100-percentile), so why 0.999 percentile? Also, care is needed when using a GEV fit for the upper tail of the distribution and record statistics, because one is unrealised and the other is realised.

We have revised this section to make things clearer. There is no mix up: records throughout the paper refer to max values. The max observed value has an empirical quantile level of 1, but when a parametric distribution is fitted, as we do, the record quantile level corresponds to $F(L)$, as explained, where F is a smooth continuous distribution. Unless the record is the upper bound (which is not impossible), record quantiles are <1 .

Concerns about the observational GEV are very valid, and addressed in greater detail now, with added uncertainty quantifications in all figures, as well as in the method section.

Figure 1 is very confusing. Why is “record quantile level” shown and what should I make out of it? Also, what kind of downscaling is used to ensure that the projected results correspond to local time series?

See above comment for the explanation of record quantile level. In addition, section 2.1 has been revised to improve clarity.

No downscaling is used, we use the gridcell values from HadEX3. We have now clarified this in the text:

“The geographical locations we refer to in Fig. 1 are located in the selected gridcells, the data shown are the gridcell values from HadEX3. ” (L160-161)

Why were the low and high quantile levels set to the approximate 10th percentile and 90th percentile of all observed record quantile levels? And how can it be justified? Also, where do 0.97 and 0.999 come from?

As we have now made more explicit in the text, the four locations in Figure 1 serve an illustrative purpose and are therefore chosen to be on opposite ends of the 2D spectrum of climate change and record extremeness. The 10th and 90th percentile are chosen to select values that are equally but oppositely extreme. The values 0.97 and 0.999 correspond to the 10th and the 90th percentile of all record quantile levels in observed gridcells. This is all explained in 2.1.

I find the **2.1 Framework** messy and hard to follow. My impression is that the authors are not up to date and are fumbling. Previous work on record-statistics is more elegant in my opinion. I wonder how many readers would follow this part and become any wiser.

We appreciate the reviewer's input and have revised section 2.1.

We do not aim to redo previous work on record statistics, but add the perspective of conditional record breaking.

Based on the poor introduction of the method together with a biased and incomplete account of past work, I'm not convinced that the results presented here provide much added value, and I lose interest in continuing putting efforts into reading and digesting them. By the way, the ERA5 reanalysis shows some suspect rainfall over parts of Africa and the analysis of record-breaking rainfall there may be biased due to poor data quality or inhomogeneities (DOI:10.1126/sciadv.ado3712).

We are sorry to hear that. We hope the revised version is less tedious. We are aware that there is a lot of methodological information in the manuscript. We feel like reducing and simplifying this to a very large degree would introduce too many possibilities for misunderstanding or misinterpretation of results. For reasons of trustworthiness and reproducibility, we therefore want to be as complete as we think is necessary. We have moved a lot of details to the methods section but the metrics that are shown in the sections after 2.1 require explanation, in our view.

References to a few additional papers on record statistics have been added where relevant. These do not contradict our results, and we do not aim to disprove or disqualify any other work in this field, we simply want to add the history-aware view on record breaking.

We acknowledge the uncertainties in ERA5. We mentioned this in the methods section, and added some additional details and caveats now:

"It has been shown that ERA5 has difficulties reproducing observed Rx1d magnitudes, and that the influence of geographical features such as coastlines and orography is not always captured. In addition, the uncertainties in the tropics are high due to low observational availability and model uncertainties. Also desert regions are sensitive to

uncertainties and errors, especially in fitted GEVs, due to the large fraction of zeros in their Rx1d samples.” (L785-789)

The term “*long record gaps*” is confusing when the discussion is about record statistics. I presume it means that there is a substantial number of missing data(?)

This refers to long periods without a record breaking event in a particular location, we have now defined this earlier in the introduction:

“Indeed, there are numerous anecdotal examples where long periods without record-breaking events or disasters (**record gaps**) plausibly led to reduced risk awareness and unpreparedness when disaster struck” (L122-123)

I find **Fig 3** very confusing presenting strange concepts such as “*cumulative record breaking probabilities conditional on the local record value from the observational period (CCP)*”. The maps don’t say much and the entire figure is a mess.

We appreciate this input. All the metrics shown in Fig 3 are defined and explained in section 2.1 and Fig 1. We are aware this section was and still is dense, but have tried to revise it to improve clarity and readability.

Fig 4 seems to be based on a misconception regarding record-statistics and randomness. It relates to the discussion about “*luck*” which I think is misplaced - rather we expect a predictable degree of variations due to randomness and probabilities, as explained in the literature on record-statistics. For instance, the statement “*a low current record is more likely to be shattered by a large margin*” is false because the magnitude of a record depends on the statistical distribution and its range varies geographically. It’s rather the length of the time series and the type of pdf that matter.

We fully agree with the reviewer: we expect randomness and variations, and because of that, there are regions that have not seen high extreme values by chance. Chance and luck could be considered synonyms; to put things in colloquial, relatable terms we refer to luck but keep the quotation marks to indicate that it is not a scientific statement.

A low current record is mathematically more likely to be shattered by a large margin. Imagine two locations with the same Rx1d climatology given by F , where F is the CDF of Rx1d. But by chance, location a’s record is 100mm and location b’s record is 70mm. In order to shatter the record by a margin of 30mm, location a needs to get >130mm rain, whereas location b only needs >100mm. The probability of getting >100mm rain is higher than the probability of >130mm. Therefore, the probability of record shattering is higher in location b with the low record.

This is exactly what we intend to convey with the paper: the actual, local current record level affects specific, local record-breaking probabilities defined as the probability of

exceeding the highest observed value. Note: the marginal record breaking probability is, by definition, **not** dependent on the local record, and indeed is dependent on the length of the record and climatology (in a non-stationary context). This remains true, and the marginal record breaking probability remains a useful statistic and diagnostic of climate change. The conditional probability is simply another metric of record-breaking probability that takes into account historical observations and approaches the issue of record-breaking from an experience-centric perspective.

The paper does not impress me and one question is what new information does this paper present beyond the findings published in the scientific literature which the authors failed to cite. It also seems to make record analytics more complicated than it needs to be. The paper is really hard to follow, and not very inspiring to read because it seems that it does not build on established knowledge regarding record-breaking statistics.

We are sorry to hear the reviewer was so unimpressed. We hope to have more clearly articulated what the conditional probability means, how it differs from marginal probabilities, and why it is a useful, additional metric that can provide impact-relevant climate information. In addition, we hope we have covered the literature that we missed earlier. Our paper builds on previous work on record-breaking done by a large variety of labs and people, and fits in a recent development of including historical observations in present and future climate risk assessments.

Reviewer #2 (Remarks to the Author):

The paper presents the analysis of record-breaking precipitation probabilities and their potential of high-impact disaster in future climates, at global scale. The analysis is developed by conditioning the record-breaking probabilities on historical observations, and considering climate change according to a large ensemble of global climate models. The disaster potential is then analyzed based on those probabilities, assuming that in case of low records and/or no recent records the preparedness for disaster risk could be lower, thus the disaster potential higher.

The topic is relevant, with novel aspects with respect to more usual analysis of changes in return levels. The results are interesting and relevant, and conclusions are supported by the analysis. I think the paper meets the scope of the journal. Anyway, I found not easy the reading, and to understand the meaning and implication of the several probabilities presented. I suggest minor revisions (listed below), mostly with the aim to enhance clarity and readability of the paper.

We thank the reviewer for the kind and constructive words, and the very helpful feedback. We are well aware that the manuscript is rather dense, and have spent considerable efforts to increase readability and streamline the narrative. See point-by-point replies below. All line numbers refer to the latexdiff file, since that makes it easier to see how the changes fit in the context.

- Section 2.1. I suggest to help the reader by linking better the textual explanations with lines in figure 1b,c, because those probabilities are better clarified and could be better understood just later in the methodology section. For example, 1) add “non stationary conditional” at line 155 after Fig1b. 2) write “conditional” cumulative record breaking probability at line 162. Also, maybe add a short explanation of what the 4 lines shows (natural variability, climate change, both, nothing). This is better explained at section 2.3, but it is not easy to understand this in section 2.1

Thanks, we have revised section 2.1. At the beginning in L149-150 we added:

“The analysis metrics refer to either marginal or conditional probabilities, in either stationary or non-stationary conditions. Marginal probabilities are history-unaware probabilities based on the mean climate, conditional probabilities are history-aware probabilities that depend on historical observations. Stationary metrics assume a counterfactual world without climate change, and non-stationary metrics include effects of anthropogenic climate changes.”

Also, we added “conditional” at former line 162, and made substantial additional changes that aim to increase straightforwardness, that are all marked in the difference file.

- Figure 1b,c: it is difficult to distinguish the 4 lines in the light yellow color. Maybe a darker one?

It was hard to see indeed, we changed the colours and they should be easier to see now.

- Line 158-159. From where do you see this? By comparing $P(Y_{nr}=n|L)$ with stationary $P_t(R|L)$?

This was unclear indeed, it was referring to the solid lines of the bottom panels of figure 1b compared to the top panels (low vs. high records). We have added more directions to the lines in the text of section 2.1 now to better guide the reader.

- Line 192. Why is MCP “moderately” sensitive to climate change, considering that it is (line 187) “only sensitive to climate change”?

In the revised text of section 2.1 this sentence is no longer there. But for completeness: MCP is indeed only sensitive to climate change as external factor changing probabilities relative to the stationary marginal reference. The definition of cumulative record-breaking probability means that MCP always increases with time (because every year there is a nonzero chance of breaking the record, even in a stationary climate). The effect of climate change on MCP emerges slowly and modifies the timeseries of MCP but not by massive margins.

- Section 2.2. at the beginning, I suggest to mention the different spatial coverage of the 3 observational datasets, to help the reader in interpreting fig2 (mostly grey for HadEX3 and REGEN).

Thanks, we mention this now in L292-295.

- Lines 262-265. How do we see this from figure 3? How to look at natural variability vs climate change is just introduced later at lines 279-285...

Non-stationary CCP includes both climate change and natural variability, and MCP only climate change. And the reviewer is right that from Fig 3 alone we could not see their separate magnitudes. We have clarified this now by referring to MCP maps in the supplementary info, which represent the global pattern of climate change in Rx1d in CMIP6 models (L375-379)

- Lines 279-285. I suggest to present the ratios with the same order they appear in figure3 (figure 3b first, then 3c). This could help the reader interpreting text together with figures.

That’s a good suggestion, we have edited this section and integrated the meaning of the ratios in the description of the figures in order of appearance now.

- Line 283. Stationary MCP?

Yes!

- Line 303. Their stationary counterpart ...is the dotted line in fig1b? add in parenthesis "(dotted)".

Yes, thanks

- Figure 3, caption for d) and e). Clearly indicate which lines show each effect (climate change, natural variability, their combination).

Good suggestion too, we've expanded the caption of Figure 3 to include this.

- Lines 461-462. What about doing for high state likelihood cells an analysis similar to that carried in this section for the low state likelihood cells?

There are not as many clusters of record breaking years for high state likelihood cells, but we have added a sentence to provide some more context:

"We also find damage reports for clusters of record-breaking events in regions without low state likelihood, such as hurricane Odile in Baja California in 2014 and the Warum floods in western Australia in March 2011. These floods were not as disastrous as the ones we addressed above, but we cannot attribute this to state likelihood given the low number of events and our simple anecdotal verification method"
(L621-626)

- Line 621. 251 years ... which is the period for the models?

This information was missing indeed. The period is historical+ssp245, so 1850-2100. We've added this in former L621. this period is only relevant for the GEV fit though, we start counting records in 1950 for models as well to match the observational period.

- Line 950 and 952 are a repetition of the same concept.

We chose to keep both sentences, as former L952 functions as summary/integration, concluding that both the PIT histograms and the reliability plots corroborate the same conclusion.

Reviewer #3 (Remarks to the Author):

Review of “Where and when will the next record-breaking precipitation disaster occur?”

This manuscript proposes a novel framework to assess the probability of future precipitation records being broken, based on the current record value and its position within a fitted climate distribution. The central idea is that regions with seemingly low historical precipitation records may, under climate change, be at disproportionately high risk of record-breaking events – a concept tied to what the authors define as “state likelihood.” The methodology is validated in model space using CMIP6 large ensembles, and then applied to real-world observational datasets using spatially pooled GEV fits.

The paper is technically ambitious, innovative, and addresses a societally relevant question. However, I have concerns about the robustness of its conclusions – especially when applied to sparse observational datasets – and about the reliance on GEV fits as a tool to infer probabilistic extremeness. These concerns are outlined below.

We thank the reviewer for the kind and encouraging words, as well as the effort invested in the review and the useful suggestions. We have tried our best to improve readability of the manuscript and add more words of caution and quantitative uncertainty quantification to the manuscript. We replied to all comments separately below. All line numbers refer to the latexdiff file, since that makes it easier to see how the changes fit in the context.

Major Comments

1. Scope and fit

The manuscript introduces a promising framework and addresses a societally relevant question with clear implications for climate risk assessment and preparedness. While the broader significance is articulated, it is embedded within a presentation that leans heavily on technical formulation and statistical diagnostics. This places the manuscript toward the more methodological end of the spectrum typically seen in Nature Communications, which prioritizes conceptual advances of broad interdisciplinary appeal. The paper could be a strong fit for the journal if some of the technical exposition were streamlined to enhance accessibility to a broader readership.

We are happy to hear this constructive view, and we agree that our manuscript is rather technical. We have found this a major challenge – achieving readability and focus on broader implications while still providing complete enough information to ensure trustworthiness and reproducibility. We have tried to put less weight on methodological details, mostly in sections 2.1 and 2.2, to focus more on the actual meaning of the results in a climate change/precipitation risk context. We refer more explicitly to the figure elements to guide the reader throughout section 2. The latexdiff file shows the

text edits we have made throughout the manuscript to try and improve readability and accessibility.

2. Overreliance on GEV fits with limited data

While I appreciate that the manuscript acknowledges many of the challenges involved in fitting GEV distributions to short observational records and makes a reasonable case for using spatial pooling to stabilize fits, I remain concerned about the degree of reliance placed on GEV-derived quantile estimates for drawing key conclusions.

A central premise of the manuscript is the use of GEV-fitted distributions to assess the quantile level of current records and estimate future record-breaking probabilities. While I appreciate the careful validation in model space and the use of spatial pooling to improve stability, I remain concerned about the degree of trust placed on GEV fits, especially when applied to relatively short observational records or pooled data. As is well known, the estimation of the shape parameter ξ and the resulting quantile behavior are highly uncertain under these conditions.

This is particularly relevant when the conclusions hinge on determining that an observed record (e.g., in 2015) is “low” – meaning that it lies at a modest quantile (e.g., 90th percentile) of the fitted distribution, and is therefore likely to be broken. I find it problematic to fit a distribution to a small sample and then use that same distribution to assess how extreme the largest value in the sample is. This risks circular reasoning, and the inferred quantile level of the record can be highly sensitive to assumptions about the tail behavior (especially when $\xi > 0$).

I encourage the authors to either:

- Quantify the uncertainty of these GEV fits explicitly (e.g., using bootstrapping or ensemble spread),
- Or temper the strength of their conclusions when applied to real-world data.

A non-parametric benchmark – such as comparing against empirical quantiles or resampling-based assessments – would go a long way in grounding the results and offering a sense of robustness.

This is a very valid point and we are happy with the more thorough uncertainty quantification that is now in the manuscript, based on the reviewer’s suggestions. The uncertainty in the quantile levels is very important indeed, as they strongly shape the results we report. We therefore focused on quantification of the uncertainty in these quantile levels, and how these uncertainties propagate into the final results. In addition we added more statements to place the results into context and emphasise the large quantitative uncertainties, e.g. in L174-185, L334-335, L387-390, and L538-539.

For explicit uncertainty quantification, we bootstrapped the fitted GEV coefficients based on their covariance matrix, described in methods subsection 4.2 (last subsection).

Using 10^4 bootstrapped GEVs, we determined the 95% confidence interval of the record quantile levels and propagated the interval's lower and upper bounds to assess the uncertainty range in the probability metrics. These ranges are shown in all figures where they are relevant (Fig. 1 And Fig. 4) and spatial representations of the uncertainty source (quantile levels) as well as propagated uncertainties are shown in mentioned method section.

We determined model-GEV uncertainties and structural model uncertainties, and find that observational GEV uncertainties and structural model uncertainties (i.e. across-model differences in the strength of climate change) are the two main sources of uncertainty, which we therefore show in the main manuscript.

We have not found a way to compare against a non-parametric alternative, since the empirical quantile level of the maximum value is 1 by definition. This means a non-parametric alternative does not distinguish between various levels of "extremeness" for the record in the historical time series. In addition, a quantile level of 1 is undefined in a GEV context whenever the shape parameter is >0 , meaning that we could not determine evolving probabilities using quantile-mapped records and model-based GEVs. Non-parametric comparisons in the model data would be more straightforward, but not insightful for observational uncertainties given model-obs differences.

By comparing the quantile level and propagated uncertainties using the parametric bootstrap approach, we hope we provide enough context.

We hope that the additional uncertainty quantification, more explicit uncertainty communication, and added emphasis on cautious interpretation add a satisfactory amount of context to ensure results are not misinterpreted or presented too strongly.

3. Inference that current records are "low"

Following on the point above, the paper's logic assumes that one can reliably assess whether a current record is "low" (i.e., not unusually high relative to the underlying or shifting distribution). But how can one assert that a record is low using the same data that generated it?

The fitted GEV distribution – often based on only ~ 66 annual values per gridcell and spatial pooling – is then used to determine that this record lies at, say, the 90th percentile. But this percentile is itself derived from the same sparse dataset, via a model with strong assumptions. It is not surprising that such fits would result in record values being labeled "moderate," particularly when the shape parameter allows for heavy tails. In many situations, I would argue the empirical distribution is more trustworthy than a fitted model with weak constraints.

In this light, the declaration that a record is "low" and that disaster potential is high seems overly confident. I strongly encourage the authors to discuss the possibility that

GEV fits may overestimate the tail behavior, and that conclusions derived from these quantile assessments should be treated with appropriate caution.

We have added a few sentences to that expresses the last point of the reviewer one-to-one in section 2.1 where we introduce the framework

“It is important to note that observational GEV distributions are subject to high uncertainty, especially in the tails. Therefore, also observed record quantile levels are uncertain, and records might erroneously be classified as particularly low or high. We quantify and show uncertainty ranges for all results, and emphasise that confidence in specific quantitative values is limited.” (L174-176)

And similar sentences at the other locations in the manuscript we listed in the reply to the previous comment.

The general statement that low records are associated with high record breaking probability is mathematically correct, however, we agree that classifying records as either low or high is shrouded in uncertainty in a practical context where limited information is available. The required caution thus applies to gridcell specific conditional probabilities based on observed records. For these values, we have emphasised that the location-specific values are subject to high uncertainty. The aggregated statistic of people exposed has large uncertainty bounds, but is more robust to local record quantile level uncertainties as the aggregation cancels errors of opposite signs.

See considerations above at point 2 as to why empirical record quantile level determination would complicate/render impossible our calculations. We require the possibility for extrapolation to unseen values that GEVs allow, yet, we agree with the reviewer that we are brushing up against the limits of extreme value theory by trying to quantify the likelihood of the most extreme even in a small sample. The GEV-uncertainty quantification described above hopefully covers part of these concerns too.

We have tried to clearly distinguish the validity of the framework (as can be tested in models) from the validity of the results based on observations, and emphasise the high level of uncertainty in observed record statistics. We hope we now paint a balanced picture that mostly focuses on proof of concept and importance of natural variability in record-breaking, without overselling deterministic, specific probability values based on observations.

4. GEV fitting under non-stationarity and compounding uncertainties

The idea of fitting time-varying GEVs to account for trends in extremes is, in principle, well motivated and aligns with current best practices in the field. However, given the

limited length of observational records (~66 years), I believe even stationary GEV fits are already subject to substantial uncertainty. Introducing time dependence in μ and σ increases model flexibility but also compounds that uncertainty – especially when applied locally. While spatial pooling helps with stability, it also imposes smoothness assumptions that may not hold in topographically or climatologically complex regions. Overall, I find the approach conceptually strong, but optimistic in its application given the data constraints. I encourage the authors to clarify how these layered uncertainties propagate into the final probability metrics (e.g., CCP, MCP, and state likelihood), particularly in the observational analysis.

We fully underwrite the concerns of the reviewer and agree that these need to be investigated to justify the approach we use. We determined observed record level quantiles based on both stationary and non-stationary GEVs fitted to observations, and find that differences are small. Yet, differences are not random: if the observed record occurred in a later year, there is a larger effect of non-stationarity of the Rx1d distribution on the quantile level. The plot below shows, per reanalysis/obs dataset, the density of differences between record quantile levels relative to non-stationary GEVs and the quantile levels of the same records relative to stationary GEVs (both with otherwise exactly the same GEV fitting procedure).

The plot shows that differences induced by GEV non-stationarity are generally very small (on the order of 0.002 or less, for quantile levels between 0.9 and 1). Yet, for later record-setting years (right), the error grows: record quantile levels determined relative to stationary GEVs are systematically biased high compared to those relative to non-stationary GEVs. This means that record quantiles determined assuming a stationary GEV would overestimate the rareness of (late) records, and thus underestimate the record-breaking probability. This behaviour is to be expected: towards the present, Rx1d distributions have shifted to higher values, thus reducing the rareness of high values. We think that including the effect of climate change on late Rx1d record quantiles is important to capture the effect of decreasing rareness of high precipitation. The small differences between stationary and non-stationary GEV-based quantile levels,

and the climate change effect that already emerges in the observations, are reason for us to stick with non-stationary GEV distributions.

Furthermore, the non-stationary fitting procedure, in theory, will only assign nonzero coefficients to the nonstationary covariates for gridcells where the smooth GMST-trend (the covariate) helps explain the variance. In theory, this means that gridcells where climate change has not emerged will be assigned their stationary GEV. In practice, the addition of the extra degrees of freedom does change the fit a bit, but the small magnitude of the quantile level differences shows that the fits are generally quite similar.

Lastly, we determined the Akaike and Bayesian information criteria (AIC and BIC) for the stationary and non-stationary fits, which weigh explained variance against model complexity. The mean AIC and BIC across all gridcells are (marginally) smaller for the nonstationary fits for all three datasets (see results below), indicating that the non-stationary fits are “better”, or at least not significantly worse, than the stationary fit for the data at hand.

```
$ dataset      : chr "HadEX3"  
$ mean AIC stat : num 3157  
$ mean AIC nonstat: num 3147  
$ mean BIC stat : num 3169  
$ mean BIC nonstat: num 3167
```

```
$ grid        : chr "REGEN"  
$ mean AIC stat : num 8500  
$ mean AIC nonstat: num 8483  
$ mean BIC stat : num 8515  
$ mean BIC nonstat: num 8508
```

```
$ grid        : chr "ERA5_rg"  
$ mean AIC stat : num 8301  
$ mean AIC nonstat: num 8258  
$ mean BIC stat : num 8317  
$ mean BIC nonstat: num 8284
```

In combination with the additional uncertainty quantification we have done for the observational GEVs and derived quantities (see replies to previous comments), we hope that the reviewer agrees with our choice for non-stationary observational GEVs. We have added a sentence to the method section to justify that choice, based on the answer here:

“Fitting non-stationary GEV distributions to observations adds uncertainty relative to stationary ones. We nonetheless choose to do this, since climate change affects Rx1d in the historical period, leading to decreasing rareness of extremes. We verified that stationary GEVs overestimate record quantile levels for records that occurred recently,

and underestimate corresponding record-breaking probabilities in the future. Differences between the two fits in earlier years are very small, indicating that the GEV fits do not become considerably different or worse due to the non-stationary fitting procedure. Lastly, Akaike and Bayesian information criteria are overall (marginally) lower for non-stationary fits for all three datasets, corroborating that non-stationary GEVs are preferred to stationary ones for the data at hand.” (L905-912)

Minor Comments

L24–26: If I’m understanding correctly, this sentence may conflate current absolute risk with the rate of increase in risk. I recommend clarifying that regions with low historical records face high absolute record-breaking probabilities, whereas regions with high records may see the sharpest relative increases in risk due to climate change.

We see the concern, and have added the word “relative” to this (revised) sentence, to indicate that it’s compared to itself without climate change. Indeed, the next sentence states what the reviewer mentions as well, and hopefully makes the difference between absolute probability and relative increase clear (L26-28).

L47–49: The statement “extreme precipitation increases approximately exponentially with global warming” could be made clearer. Which aspect of precipitation extremes increase exponentially with global warming? Is it intensity (weakly nonlinear) or frequency (highly nonlinear and depending exponentially on the local Clausius-Clapeyron relationship; e.g., see Martinez-Villalobos & Neelin 2023, equation 3 or 15, <https://www.nature.com/articles/s41598-023-32372-3>). Please clarify.

Here we simply mean to refer to the simplified clausius-clapeyron relation, that prescribes a 7% increase per degree of warming. It is approximate, since 1. the unsimplified CC-relation is not constant with temperature and 2. Other factors impact the actual increase of extreme precipitation as well, CC does not cover the full response. Regardless of what we intended to say, we understand that this statement can lead to confusion so we changed the sentence to: “Theory, models and observations agree that, on average, extreme precipitation increases at a rate of approximately 7% per degree Celsius of global warming.” (L55-56)

L128: The GEV framework includes several layers of uncertainty, especially when fitting time-dependent parameters. How are these propagated into the final probability estimates? (see my major comments)

See our answers to the major comments :)

L137–138: The illustrative regions shown in figures are interesting, but how geographically representative are they? Were these selected a priori, or because they yielded good signal?

We describe the selection procedure in the first paragraph of section 2.1– and updated it to hopefully be more clear. The regions were selected in a “data-driven” manner; we subsetted based on low/high quantile level and low/high climate change, and in the intersections of these subsets we picked points that were in different parts of the globe. The signal strength was not taken into consideration, the final selection within the subsets was basically random and only based on geographical location. Given the subsetting based on record level and climate change, all gridcells within a subset would yield similar signals.

L217–220: Attribution to localized convective systems may be an oversimplification – such storms are often embedded in broader synoptic patterns. Please consider revising.

We agree that was a bit simplistic, we have made the sentence more general – we hope the reviewer agrees that the phrasing below applies, no matter the exact system or its size. The point we’re trying to convey is that record precipitation rates do not cover a multi-gridcell area.

“In addition, because precipitating systems exhibit strong spatial gradients at relatively small scales, record precipitation rates do not affect areas spanning multiple observed gridcells.” (L310-311)

L262–268: How does internal variability influence the results? For instance, are different ensemble members consistent in identifying “low-record” locations?

No, different members have uncorrelated record quantile level patterns, exactly because of natural variability. Averaged over a large ensemble, the record quantile level in each grid cell converges to the marginal one ($1 - (1/(1+N))$), but in individual members the local deviations from this mean rate are uncorrelated.

Any pattern that is consistent across members must be related to some forcing that is common across members,, whereas anything resulting from natural variability should be random. Any systematic consistency in propagated member record quantile level patterns (i.e. $F_t(L_m)$) is a result of climate change acting to generally dampen record quantiles.

L351–352: “The sections above showed...” seems too strong. Much of the analysis supports the idea that conditioning on local historical records may be important, but this is more a hypothesis than a conclusion fully demonstrated.

The statement in this sentence is true from a mathematical perspective: conditioning on the historical record affects record-breaking probability calculations.

Whether it is also useful or applicable in a real world context is another question, and we see how the sentence might be suggestive in this way, and have modified it to the more neutral:

“The sections above showed that conditioning on local historical records **changes** natural hazard probability quantification” (L497-498)

L364–366: The sentence implies a functional relationship between state likelihood and the discrepancy between actual risk and preparedness. It appears that state likelihood is being used as a statistical proxy for societal awareness or preparedness – a creative and potentially useful framing. However, while the narrative about awareness biases and lack of experience is plausible and well discussed, this connection remains conceptual and is not quantitatively evaluated in the manuscript. Is a linear or threshold-based relationship assumed? If not, I suggest clarifying that this is an interpretive use of the metric, not a demonstrated empirical link.

The reviewer is absolutely right in highlighting that we do not present a quantitatively verified (or verifiable) model of disaster preparedness. We clarify this now in line “we emphasise that our framework relating state likelihood and disaster preparedness is an interpretive, conceptual framework and does not provide a quantitative, empirical threshold beyond which risks measurably increase.” (L626-628)

This is an interesting and timely paper that introduces a creative approach to diagnosing regional vulnerability to record-breaking precipitation extremes. The use of model ensembles for method validation is well executed, and the concept of state likelihood is clearly defined and tested. However, the application to observational data, and the inference that current records are “low” based on fitted GEV distributions, raise statistical concerns that warrant further attention. The conclusions, while potentially impactful, should be more cautiously framed, and uncertainties more clearly communicated. With these revisions, the manuscript could make a valuable contribution to the climate extremes literature.

Thanks again for the kind and constructive words, we are grateful for the suggestions and feel they improved the manuscript a lot.

Dear reviewers,

Thanks very much for the positive review of our revised work. We have included the proper working link to the data and code repository, which now includes all final code and data corresponding to this manuscript version.

Thanks very much,

Iris de Vries, on behalf of all authors.

**Review of ‘Where and when will the next record-breaking precipitation disaster occur?’
by de Vries *et al.***

In general, I find this manuscript a bit disappointing and suggest rejection. Below are some comments on which my evaluation is based.

I found the manuscript a bit disappointing in terms of acknowledging past research on record-breaking events, and it's important to account for relevant work and publications to give the reader some assurance that the authors are so well informed of the topic that it's worth spending the effort to read the paper. Furthermore, in science it's expected that colleagues are recognised for their work, but a lack of due literary research may also potentially indicate a biased presentation. I must admit that I sometimes wonder if my peers actually read the literature, as it's quite typical that relevant papers are not cited but a few are cited multiple times from within a smaller part of the community. At least, it's not a bad idea to do a thorough literary research and provide a comprehensive account on the topic of record-breaking statistics within climate research to avoid glaringly different stories on record-breaking precipitation (... and if they do differ, then the differences should be explained). Also, some journals limit the number of self-citations to a small number of studies.

The statement “*The high spatiotemporal variability of precipitation implies that events that are much more extreme than any previously observed event can occur, catching society by surprise*” may not be correct, as explained in DOI:10.1126/sciadv.ado3712, which showed that there have been more record-breaking temperatures (i.e. more extremes) in the tropics with smoother variations than in the high-latitude with more abrupt and pronounced variations. Also, this statement neglects the fact that the duration of the streaks with near-record-breaking weather may be just as damaging as the magnitude of more ephemeral events.

The sentence “*The high spatiotemporal variability of precipitation implies that events that are much more extreme than any previously observed event can occur, catching society by surprise*” can also be a bit misleading because spatial/geographical variations of weather statistics are often well-known and highly predictable. It is mostly the temporal variability that provides surprises.

The sentence “*Extreme precipitation increases approximately exponentially with global warming, with a rate of 7% per degree Celsius of global warming [3, 13, 18, 24], but more extreme events see relatively stronger increases [12, 14, 25, 26]*” and the whole paragraph ignores studies which indicate that also the fraction of Earth's surface area receiving daily rainfall is a factor determining the extremity of precipitation (see e.g. DOI:10.1038/s41612-024-00794-z and references therein). I think it's important to be up-to-date regarding the science in order to convince the readers, and hence, it's important to acknowledge also recent progress, even if it's an effort to keep abreast.

The passage “*These so-called record-shattering events [22, 23] – large jumps in record values – are rare, but can occur in the absence of strong climate change, due to natural variability*”

alone” this is only true for long time series, but not so for short series when it’s more typical with large jumps in record-breaking levels. Also, it really depends on the underlying type of distribution of the variable, and we expect different step changes if it were normally distributed or followed some gamma distribution. Rather than citing their own work in this paragraph, they could provide a broader account of record-statistics such as the one in DOI:10.1126/sciadv.ado3712.

I don’t understand the point made by this sentence: “*Marginal record-breaking probabilities can be assessed by averaging many climate realisations [22, 23, 27], but local, real-world record breaking probabilities are dependent on the locally observed current record*” and I’m not sure that the statement is correct. Record-breaking probabilities usually are not estimated through taking averages and they vary from place to place and with the number of measurements taken. This is explained in literature not cited by de Vries *et al.* Also, future record breaking probabilities have been addressed before and there are published papers on the same topic as presented in this manuscript.

What’s the justification for the following statement: “*The observed local historical Rx1d record is associated with a certain quantile (percentile) level in the local Rx1d distribution, which defines the instantaneous record breaking probability*” and how do certain percentiles define record-breaking probability? And why call it “instantaneous”? Record-breaking probability is always defined by a collection of data and changes over time in a chronological fashion, as discussed previously in the literature.

The sentence “*For four single gridcells with all possible combinations of low/high record quantile level and weak/strong climate change, we show an example of the analysis metrics in Fig. 1.*” is confusing. What is meant by “*low/high record quantile level*”. It seems like records (measurements) are mixed up with records (highest or lowest recorded value). Isn’t a record the same as q_1 (100-percentile), so why 0.999 percentile? Also, care is needed when using a GEV fit for the upper tail of the distribution and record statistics, because one is unrealised and the other is realised. **Figure 1** is very confusing. Why is “record quantile level” shown and what should I make out of it? Also, what kind of downscaling is used to ensure that the projected results correspond to local time series?

Why were the low and high quantile levels set to the approximate 10th percentile and 90th percentile of all observed record quantile levels? And how can it be justified? Also, where do 0.97 and 0.999 come from?

I find the **2.1 Framework** messy and hard to follow. My impression is that the authors are not up to date and are fumbling. Previous work on record-statistics is more elegant in my opinion. I wonder how many readers would follow this part and become any wiser.

Based on the poor introduction of the method together with a biased and incomplete account of past work, I’m not convinced that the results presented here provide much added value, and I lose interest in continuing putting efforts into reading and digesting them. By the way, the ERA5

reanalysis shows some suspect rainfall over parts of Africa and the analysis of record-breaking rainfall there may be biased due to poor data quality or inhomogeneities (DOI:10.1126/sciadv.ado3712).

The term "*long record gaps*" is confusing when the discussion is about record statistics. I presume it means that there is a substantial number of missing data(?)

I find **Fig 3** very confusing presenting strange concepts such as "*cumulative record breaking probabilities conditional on the local record value from the observational period (CCP)*". The maps don't say much and the entire figure is a mess.

Fig 4 seems to be based on a misconception regarding record-statistics and randomness. It relates to the discussion about "*luck*" which I think is misplaced - rather we expect a predictable degree of variations due to randomness and probabilities, as explained in the literature on record-statistics. For instance, the statement "*a low current record is more likely to be shattered by a large margin*" is false because the magnitude of a record depends on the statistical distribution and its range varies geographically. It's rather the length of the time series and the type of pdf that matter.

The paper does not impress me and one question is what new information does this paper present beyond the findings published in the scientific literature which the authors failed to cite. It also seems to make record analytics more complicated than it needs to be. The paper is really hard to follow, and not very inspiring to read because it seems that it does not build on established knowledge regarding record-breaking statistics.